# SGLT2 Inhibitors: From Structure–Effect Relationship to Pharmacological Response

**DOI:** 10.3390/ijms26146937

**Published:** 2025-07-19

**Authors:** Teodora Mateoc, Andrei-Luca Dumitrascu, Corina Flangea, Daniela Puscasiu, Tania Vlad, Roxana Popescu, Cristina Marina, Daliborca-Cristina Vlad

**Affiliations:** 1Doctoral School, Faculty of Medicine, “Victor Babeș” University of Medicine and Pharmacy, 2nd Eftimie Murgu Square, 300041 Timisoara, Romania; teodora.mateoc@umft.ro; 2Intensive Care Unit Department, “Pius Brinzeu” County Emergency Hospital, Liviu Rebreanu Blvd. 156, 300723 Timisoara, Romania; al.dumitrascu00@gmail.com; 3Department of Biochemistry and Pharmacology, Faculty of Medicine, “Victor Babeș” University of Medicine and Pharmacy, 2nd Eftimie Murgu Square, 300041 Timisoara, Romania; cristina.marina@umft.ro (C.M.); vlad.daliborca@umft.ro (D.-C.V.); 4ANAPATMOL Research Center, “Victor Babeș” University of Medicine and Pharmacy of Timisoara, 2nd Eftimie Murgu, 300041 Timisoara, Romania; popescu.roxana@umft.ro; 5Department of Cell and Molecular Biology, Faculty of Medicine, “Victor Babeș” University of Medicine and Pharmacy, 2nd Eftimie Murgu Square, 300041 Timisoara, Romania; tania.vlad@umft.ro

**Keywords:** SGLT2 inhibitors, metabolic modulation, lipid accumulation, pharmacological response

## Abstract

SGLT2 inhibitors have become increasingly used due to their effectiveness in improving not only type 2 diabetes but also cardiovascular, renal and hepatic diseases, as well as the obesity found in metabolic syndrome. Starting from the structure of gliflozins, modifications of the carbohydrate part, aglycone, and also the glycosidic bond between them can determine variations in pharmacokinetic and pharmacodynamic properties. SGLT2 inhibitors, in addition to reducing blood glucose levels, improve alterations in lipid metabolism by diverting excessively accumulated lipids in tissues towards mobilization, lipolysis, β-oxidation, ketogenesis and the utilization of ketone bodies. This enhances anti-inflammatory properties by decreasing the levels of some proinflammatory mediators and by modulating some cell signaling pathways. Thus, in this review, the intimate mechanisms by which SGLT2 inhibitors achieve these therapeutic effects in the various conditions belonging to metabolic syndrome and beyond were described, along with the structure–effect relationship with some specific features of each gliflozin. Starting from these findings, further modeling of these molecules may lead to the creation of new therapeutic uses. Further research is needed to broaden the range of indications and also eliminate adverse effects, such as phenomena leading to lower limb amputations.

## 1. Introduction

Chronic hyperglycemia, characteristic of diabetes mellitus (DM), produces a series of complications through prolonged exposure to high glucose concentrations, leading to slow, non-enzymatic glycosylation over time. These structural changes lead to the appearance of irreversible complications such as the following: diabetic retinopathy, diabetic nephropathy, diabetic neuropathy and cardiovascular damage [1,2,3]. But an increase in blood glucose can also be induced by other factors, such as stress, obesity and some anti-hypertensive drugs, such as diuretics, favoring the appearance of type 2 DM (DM2) and creating a vicious circle [4,5,6].

According to the International Diabetes Federation (IDF), there is a growing trend in the prevalence of DM, with 537 million cases estimated in 2021 and a forecast of prevalence increasing to 783 million by 2045 [7,8,9]. In particular, based on statistical data provided by the IDF, diabetes is expected to affect 643 million people by 2030 [10]. Furthermore, it is estimated that of all patients with DM, 96% have DM2 [11]. This related information suggests that diabetes has reached pandemic proportions, impacting over half a billion people worldwide [7]. China remains the country with the highest diabetes rate, with approximately 140 million diabetic patients aged between 20 and 79 years old [12]. In a 2024 report, the United States Centers for Disease Control showed significant statistics indicating that 14,7% of the adult population suffers from diabetes [13]. One of the biggest concerns in the medical field is undiagnosed diabetes. Several studies reveal that approximately 50% of all people with diabetes are undiagnosed [14]. The healthcare expenses associated with diabetes treatment are significantly higher than those for patients without diabetes [15]. The estimated diabetic-related costs for 2021 were USD 996 billion and are projected to reach USD 1054 billion by 2045, with a major socioeconomic impact [7].

Recent KDIGO guidelines [16,17] have highlighted SGLT2 inhibitors as the first therapeutic option for patients with DM2 and chronical kidney disease (CKD), alongside metformin, renin–angiotensin–aldosterone system (RAAS) blockers and statins [18], with the main concern being related to the fact that one in three people with DM also has CKD [19]. Furthermore, the American Diabetes Association (ADA) and European Association for the Study of Diabetes (EASD) recommend SGLT2 inhibitors early in the management of patients with DM2 [20]. In the last few years, another promising medication, named finerenone, entered the medical market. It was found by at least two clinical trials, FIDELIO-DKD and FIGARO-DKD, that finerenone, a nonsteroidal selective mineralocorticoid receptor antagonist, reduced cardiorenal risk [21,22]. Finerenone was authorized by the FDA in 2021, and European Society of Cardiology guidelines recommend considering using the pharmaceutical agent in the treatment of heart failure with mildly reduced ejection fraction [23]. In patients with DM2, the main cause of mortality is represented by cardiovascular diseases (CVDs), especially atherosclerotic/ischemic diseases occurring in the context of hypertension, dyslipidemia and obesity associated with constantly elevated blood glucose levels [24]. Insulin resistance is considered to be an independent risk factor for the development of CVD [25]. In general, with a few exceptions, most patients with DM2 or pre-diabetes, as indicated by the RAAS, present not only elevated blood glucose levels but also a group of alterations including obesity, hypertension, hepatic steatosis and other pathological changes [26]. Special attention in DM2 is paid to obesity, the incidence of which has tripled in the last 50 years [15]. This condition maintains the lesions of DM2 but also favors and aggravates CVD. It seems that the fight against obesity is harder than it seems at first glance, with surgical treatments such as bariatric surgery, sleeve gastrectomy and Roux-en-Y gastric bypass being increasingly used today [27].

In this context, it is important to identify drugs that act not only on blood glucose but also on other links involved in this vicious circle. SGLT inhibitors are compounds that have demonstrated clinical efficacy, but there are some aspects, some beneficial, others undesirable, whose mechanisms of action are partially elucidated. In this review, we want to discuss current controversies related to the use of SGLT2 inhibitors, trying to promote hypotheses about some mechanisms that are still incompletely elucidated. To this end, we discuss studies related to their mechanism of action in relation to their chemical structure in order to correlate existing data and explain some empirically observed beneficial therapeutic effects and also some adverse effects.

## 2. Methodology

The literature search strategy was carried out in the PubMed, Google Scholar, ScienceDirect and Web of Science databases, giving priority to the newest articles in the field. The time interval was generally extended to 10 years; sometimes, when we considered it relevant, articles older by 1 or 2 years compared to the proposed period were also taken into consideration (for example, those with historical value referring to certain discoveries). Meeting abstracts and Proceedings, Letters to the Editor, Commentaries and Editorials were excluded; only peer-reviewed articles were taken into account. The main keywords were “SGLT2 inhibitors”, “SGLT1”, “SGLT2”, “SGLT3”, “SGLT4, “SGLT5”, “gliflozins”, “O-glucoside gliflozins”, “C-glucoside gliflozins” and combinations between them. Combinations between “SGLT2 inhibitors” and “euglycemic diabetic ketoacidosis”, “diabetic cardiomyopathy”, “arrhythmias”, “coronary artery disease”, “non-alcoholic fatty liver disease”, “renal disease” and “obesity” were also used. Last but not least, we used multiple combinations of keywords listed above. Sometimes, searching for certain information led to new directions, ultimately resulting in the section dedicated to the off-label use of gliflozins. As these are drugs with molecular modeling potential open to new research, we emphasized the known structure–effect relationships as well as the main mechanisms by which SGLT2 inhibitors exert their therapeutic benefits and adverse reactions, because minor structural changes can sometimes produce major transformations.

## 3. SGLTs Seen as Therapeutic Targets

Sodium glucose cotransporters (SGLTs) are a family of membrane-localized proteins specialized in the active reabsorption of sodium-coupled carbohydrates against their concentration gradient [28,29]. The SGLT protein family contains six known representatives so far, and monosaccharides are preferentially absorbed with their contribution. Hence, SGLT type 1 (SGLT1) transports glucose and galactose; SGLT type 2 (SGLT2) transports only glucose; type 3 (SGLT3) works as a glucose sensor; type 4 and type 5 (SGLT4 and SGLT5, respectively) recognize fructose and mannose; SGLT type 6 (SGLT6) intervenes in the absorption of myo-inositol [30]. Among them, the most studied are SGLT1 and SGLT2, targeted by recently introduced therapeutic compounds with inhibitory action. These transporters bind both sodium and monosaccharides at the same binding site [30]. Thus, SGLT1 transports a monosaccharide molecule coupled with one sodium ion, while SGLT2 transports glucose coupled with two sodium ions [31].

### 3.1. SGLT1

SGLT1 is found at the intestinal brush border and is responsible for the absorption of exogenous glucose and galactose [29]. In addition to the main intestinal location, other locations such as the heart, brain, skeletal muscle and kidney have also been reported [32]. In the intestine, its expression is regulated by dietary carbohydrate intake, increasing in high-carbohydrate diets. Furthermore, it exhibits circadian variations with a maximum activity during the day [29,33]. There is a direct relationship between SGLT1 and the release of intestinal hormones such as glucagon-like peptide-1 and glucose-dependent insulinotropic peptide [34]. The loss of functionality of this receptor through various mutations results in a syndrome of glucose and galactose malabsorption [35].

### 3.2. SGLT2

SGLT2 is found only in the kidneys, at the proximal level, and plays a role in the reabsorption of glomerularly filtered glucose [36], being responsible for 90–95% of reabsorbed glucose [37]. Mutations in this receptor lead to an inability to reabsorb glucose, manifested as renal diabetes [35]. On the other hand, SGLT2 is overexpressed in hyperglycemia states associated with hyperinsulinemia, a situation observed in DM2, being responsible for glucose overload. Along with glucose, in patients with DM2, there is an increase in sodium reabsorption with a tubular overload and additional stimulation of the RAAS. Vasoconstriction resulting from increased production of angiotensin 2 could increase intraglomerular pressure, producing alterations at this level [37].

### 3.3. SGLT3

SGLT3 is localized in the intestinal epithelium, in myenteric neurons, and in skeletal muscle at the neuromuscular junction. Other locations include the portal vein and the kidney [38]. SGLT3 appears to play a role as a glucose sensor, with its expression being reduced in obese patients [39]. Upon the performance of a Roux-en-Y gastric bypass, a common surgical procedure in obese patients, especially those with diabetes, its expression seems to become similar to that of non-obese subjects [39,40].

### 3.4. SGLT4

SGLT4 transports mannose and fructose, being involved in the reabsorption of these sugars. It is localized in the epithelium of the proximal tubules [41] and also in the mucosa of the small intestine [42]. It has been observed that a diet rich in fructose has a stimulatory effect on fructose reabsorption by increasing the expression of SGLT4 [41,43].

### 3.5. SGLT5

SGLT5 is a monosaccharide transporter located at the apical pole of renal tubular epithelial cells [44]. It is involved in the tubular reabsorption of 1-deoxyglucose, also called 1,5-anhydroglucitol (1,5-AG), a carbohydrate considered to be the most abundant polyol in the blood [45]. In addition to 1,5-AG, SGLT5 can transport fructose and mannose, but the affinity for mannose is 3.5 times lower than that of 1,5-AG [44,46]. In fact, SGLT5 is involved in the preferential transport of monosaccharides that do not possess an OH group at the C1 carbon atom and fructose with an OH group at C2. Thus, SGLT5 is a preferential transporter of hexoses that do not exhibit an OH group at C1 and instead possess one at C2, with the same orientation as that of mannose [46]. Related to this transporter, a diet high in fructose may increase blood pressure due to intensification of sodium reabsorption through upregulation of SGLT5 [47].

### 3.6. SGLT6

SGLT6 is a transporter for glucose and myo-inositol with high expression in the brain, especially in the hypothalamus and substantia nigra areas involved in satiety and reward systems. In the intestine, its expression has been observed in myenteric plexus and enterocytes [48]. It is also found in the kidney, with a low affinity for glucose [49]. Proper functioning of the membrane flux provided by SGLT6 offers protection against cerebral edema in ischemic stroke [50].

Drug modulation of these receptors may bring therapeutic benefits in certain yet unexplored pathologies but may also be responsible for the production of adverse reactions. The combined action of these receptors in stimulatory or inhibitory senses may lead to the discovery of new compounds or the exploration of existing ones.

## 4. Structural Aspects of Compounds with Inhibitory Action on SGLTs

In general, the carbohydrate part is responsible for the inhibitory action on SGLTs through glycomimetic action [51]. The identification of structures capable of inhibiting SGLTs started from the observation that the natural compound Phlorizin achieved non-selective inhibition of these, causing glycosuria and a decrease in blood glucose [52]. Structurally, Phlorizin contains a carbohydrate moiety (β-glucose) O-glycosidically linked to a bicyclic flavonoid component of the dihydrochalcone type [53] (Figure 1). Starting from the O-glycosidic structure of Phlorizin, other structures were developed. The first introduced in therapy were Dapagliflozin and Canagliflozin, quickly followed by Empagliflozin and Ertugliflozin, paving the way for a new class of compounds: the gliflozins [52,54].

### 4.1. O-Glucoside Gliflozins

The most therapeutically explored O-glycosidic gliflozins are Sergliflozin and Remogliflozin (Figure 1). The replacement of the propanone group in the Phlorizin structure with a methylene group resulted in Sergliflozin, increasing SGLT2 selectivity [55]. Although the SGLT2 inhibitory effect of Sergliflozin was effective, its short half-life of 1–1.5 h made it unusable as a therapeutic alternative [56].

Remogliflozin has a modified aglycone with both a pyrazole ring and isopropyl substituents, which improves its selective activity on SGLT2 [55]. Although Remogliflozin has a short half-life at 1.39 h [57], in a review study comparing it with other gliflozins, it showed a significant effect on weight loss and an increase in HDL, but was inferior in terms of reducing postprandial glycemia [58]. Remogliflozin is currently withdrawn from the United States market [57].

One of the essential problems is represented by the hydrolysis of the O-glycosidic bond in the intestinal tract by β-glycosidases. To reduce the affinity of these enzymes for O-glycosidic bonds, O-gliflozin has been administered orally as a prodrug (etabonate esters), which is further hydrolyzed to release the active substance [55]. The essential role of these structural modifications opens the modeling of molecules to develop more effective gliflozin class drugs with improved pharmacokinetics.

### 4.2. C-Glucoside Gliflozins

Unlike Phlorizin, almost all gliflozins are C-glucosides. The C-C bond allows oral administration and prevents the hydrolysis of the glycosidic bond in the gastrointestinal tract, making them stable molecules, resistant to intestinal degradation [59]. The most well-known C-gliflozins currently are Canagliflozin, Dapagliflozin, Empagliflozin, Ipragliflozin, Bexagliflozin (Figure 2) and also structures with a modified glucidic moiety such as Ertugliflozin, Luseogliflozin, Sotagliflozin, Henagliflozin and Tofogliflozin (Figure 3).

In an investigation of the structure–pharmacodynamic effect relationship, it was observed that Dapagliflozin binds NO if there is a donor, with binding being performed at the alkyl radical involved in the ether bond [59]. This aspect adds a favorable effect in the case of CVD accompanied by ischemia. Without the methylene bridge, this effect is not observed [60]. Replacing the chlorine atom in the Dapagliflozin molecule with a methyl group results in a loss of activity. Action is improved by the introduction of a short alkyl substituent in the para position of the distal phenyl moiety [60]. Replacing the distal phenyl ring with thiophene as in the case of Canagliflozin will increase potency and selectivity for SGLT2 [61,62]. Ipragliflozin, which contains a benzothiophene group instead of distal phenyl, has lower potency but increased selectivity for SGLT2 [55,61]. Therefore, it can be said that distal sulfur-containing rings contribute to increased activity and selectivity for SGLT2, while a long chain attached to distal phenyl reduces selectivity for SGLT1 [61]. It seems that Bexagliflozin, derived from Dapagliflozin, due to the addition of cyclopropyloxyethoxy group, acts as an inhibitor for both SGLT1 and SGLT2 [61]. In general, any change in the substituents of the distal phenyl residue leads to a change in the inhibitory potential on SGLT2.

From a lipophilicity point of view, the presence of a fluorine atom in the Canagliflozin structure increases solubility compared to that of Dapagliflozin, while Empagliflozin has a lower lipophilicity due to the furan-type ring [63]. Thus, gliflozins with higher lipophilicity will display a lower binding to plasma proteins [63], a higher potency and also a higher selectivity for SGLT2, as observed in the case of Canagliflozin [55].

Ertugliflozin and Henagliflozin have similar structures with a dioxa-bicyclo-octane modification at the sugar moiety, a modification associated with increased potency and activity in terms of SGLT2 inhibition [55,64,65,66]. Another advantage is a long half-life of 17 h for Ertugliflozin [55,67] and 15 h for Henagliflozin [68], allowing once-daily administration. A long half-life may be achieved by slow dissociation from the SGLT binding site caused by the modification of carbohydrate structure. Sotagliflozin is structurally similar to Ertugliflozin and Dapagliflozin, the only difference being the modifications to the sugar moiety, where the -CH_2_OH group at position 6 of glucose is replaced by a thio-methyl group. This modified structure has an inhibitory effect on both SGLT2 and SGLT1 [55,69]. Sotagliflozin appears to have significant inhibition of intestinal SGLT1 with minimal effect on renal SGLT1 [70,71]. Moreover, recent studies have demonstrated that inhibition of both SGLT1 and SGLT2 has additional therapeutic benefits in heart failure, when SGLT1 is overexpressed [72,73]. The overexpression of SGLT1 by cardiomyocytes is associated with oxidative stress, myocardial hypertrophy and fibrosis [73]. SGLT1 inhibition may also influence the gut microbiota. Blocking SGLT1 in the gut prevents glucose absorption and promotes intestinal fermentation, reducing dysbiosis in favor of beneficial bacteria and increasing the production of short-chain fatty acids. The advantage may also be directed to the heart, as dysbiosis is known to be associated with cardiovascular and renal disease [74]. Luseogliflozin exhibits structural modifications at both the carbohydrate and aglycone moieties. Luseogliflozin contains a 5-thioglucose ring, which confers the property of binding tightly to SGLT2 and dissociating very slowly compared to glucose-containing SGLT2 inhibitors [75]. The effect on SGLT2 is intense and selective, producing glycosuria that is maintained for a period of 48 h after a single dose [55]. Tofogliflozin, first described by Ohtake et al. [76], is not a substance with a real glucose modification, but the appearance of an O-spiroketal structure gives the molecule a greater selectivity for SGLT2. Tofogliflozin is one of the most selective SGLT2 inhibitors [77] but has a short half-life of 5–6 h [78]. The O-spiroketal-C-glucoside-type structure also improves bioavailability, while the CH_3_-CH_2_- group attached to the distal benzyl of the aglycone increases the lipophilicity of the molecule [79].

It can be seen that small structural changes can produce new properties in terms of improving and modulating pharmacokinetic and pharmacodynamic aspects. In general, changes in glucose structure cause long-term attachment to SGLT2 and the addition of hydrophobic chains to the aglycone, and O-spiroketal structures increase selectivity and lipophilicity.

## 5. Studies That Explain the Global Therapeutic Action of SGLT Inhibitors

The effects of SGLT inhibitors are not limited to reducing blood glucose levels, a desired effect in DM2; they also include additional beneficial therapeutic consequences and adverse reactions. Below, we present the main mechanisms for which this category of substances has so many therapeutic indications, and we try to bring to light arguments for and against their use in certain questionable situations.

### 5.1. DM2 and Beyond

The well-known general mechanism of SGLT inhibitors is the inhibition of intracellular glucose uptake both in the kidney and in other tissues and organs. At the renal level, the prevention of glucose reabsorption will result in increased glucose elimination. The inhibition of intestinal glucose absorption brought into the body by exogenous input also contributes to lowering blood glucose and reducing caloric intake [80,81,82,83]. In addition to global glycemic control, reducing HbA1c and decreasing tissue exposure to high blood glucose levels, an improvement in insulin resistance can also be observed [84,85,86,87]. However, as a general effect of SGLT inhibitors on blood glucose homeostasis, a reduction in the body’s exposure to glucose can be noted by promoting its elimination and also by inhibiting absorption. Due to this observation, SGLT inhibitors demonstrate their usefulness as adjuncts in DM1 along with insulin administration [88,89].

One of the prices paid for the effectiveness of SGLT inhibitors in DM is euglycemic diabetic ketoacidosis (EDKA). Once the increase in urinary glucose excretion and, if necessary, the reduction in intestinal absorption, glucagon production is stimulated under conditions of hypoinsulinemia. Glucagon will stimulate gluconeogenesis, lipolysis and ketogenesis [90]. The increase in glucagon/insulin ratio in favor of glucagon enhances lipolysis with massive release of free fatty acids and synthesis of ketone bodies [91,92]. The lower the insulin production, the greater the risk of lipolysis and ketogenesis. Volume depletion combined with hypoinsulinemia is the triggering factor for ketoacidosis, activating the compensatory release of catecholamines and glucocorticoids with an even more pronounced increase in lipolysis in adipose tissue [93,94,95]. For this reason, in DM1, due to the absolute insulin deficiency with the compensation described above, the FDA does not recommend the use of SGLT2 inhibitors due to the major risk of producing EDKA, while the European Commission has allowed the use of Dapagliflozin and Sotagliflozin in patients with DM1 and a body mass index of over 27 kg/m^2^ since 2019 [96,97]. Through the osmotic diuresis they produce, SGLT2 inhibitors will cause hypovolemia that accentuates dehydration from ketoacidosis by stimulating the release of cortisol and adrenaline that deepens lipolysis and ketogenesis [98]. The reduction in circulating volume will also have an activating effect on β1 adrenergic receptors, further increasing glucagon production [99]. The administration of SGLT2 inhibitors will increase glycosuria with a decrease to normal values of blood glycemia, making EDKA difficult to diagnose [92] (Figure 4). In addition, at low blood glucose levels, the need for insulin cannot be accurately assessed [100]. In general, dehydration and hypoinsulinemia, each alone, are necessary but not sufficient to produce acidosis, and together they can generate EDKA. When insulinopenic rats are treated with Dapagliflozin, a 70% increase in adipose tissue lipolysis and ketogenesis was observed, producing EDKA [93]. Even though EDKA is a rare adverse effect, it has been described. In some studies, EDKA was an event noticed only in diabetic patients treated with SGLT2 inhibitors [101]; in fact, in a group of patients receiving gliflozins, 56.3% suffered an episode of EDKA, whereas in those receiving other antidiabetic treatments, the incidence was 2.6% [102]. On the other hand, EDKA is 3.7 times more likely to occur during SGLT2 inhibitor treatment compared to other drug classes [103]. Because EDKA is a life-threatening condition, SGLT2 inhibitor therapy should be discontinued once it occurs [104]. In the event of an episode of EDKA, it is not recommended to restart the treatment with SGLT2 inhibitors after the patient has been stabilized because there is an increased risk of recurrence [98].

The factors that enhance EDKA appearance in the context of the use of SGLT2 inhibitors are represented by processes that enhance ketone body synthesis (anorexia, ketogenic diet), dehydration (alcohol consumption or diuretic drugs), and processes that increase oxidative stress and cause discharges of catecholamines and corticosteroids (infections, postoperative and intraoperative stress, stroke) [98,105,106,107]. Recent case reports showed EDKA in patients treated with gliflozins in different contexts associated with the factors described above. An 83-year-old patient with DM2 treated with empagliflozin was admitted to the emergency department with intraparenchymal cerebral hemorrhage in the left occipital lobe attributed to uncontrolled high blood pressure. The patient presented with metabolic acidosis but without dehydration. The authors believe that EDKA occurred in the context of oxidative stress and the proinflammatory status generated by vascular and cerebral lesions that led to an imbalance in the glucagon/insulin ratio [108]. Another case of EDKA was in a 90-year-old patient with DM2 also treated with empagliflozin who presented with a urinary tract infection associated with anionic gap metabolic acidosis and with a history of ischemic cardiomyopathy and multiple myeloma in remission. Here, the anionic gap persisted 2 days after the start of insulin therapy [109]. Three other cases of EDKA in adult patients (68, 66 and 55 years old) with type 2 diabetes treated with SGLT2 inhibitors have been reported. Postoperative stress and lack of food intake have played an important role in the generation of ketoacidosis [110]. Moreover, other authors have described interesting cases in patients with DM2 that included an SGLT2 inhibitor in the therapeutic regimen, patients diagnosed with EDKA in the context of food deprivation [111], and infections such as pulmonary infections [112], genital abscesses [113], sepsis [114], and an intraoperative infection in the case of a craniotomy performed for tumor resection [115]. The precipitating factor that contributed to the development of EDKA was prolonged anorexia for two consecutive days of a vertebral and rib fracture. C-peptide values were within normal limits, excluding damage to β pancreatic cells. Here, starvation generated an increased synthesis of ketone bodies in the context of glucose elimination, which had an important contribution even though insulin secretion was normal [116].

Evidence of EDKA occurring after discontinuation of Empagliflozin treatment in a 45-year-old male patient with DM2 in the context of starting a ketogenic diet has been published [117]. Another case report described a 54-year-old woman with DM2 where EDKA occurred in the context of insulin discontinuation, a low-carbohydrate diet, prolonged fasting, alcohol consumption, and SARS-CoV-2 infection [118]. However, it seems that a ketogenic diet, whether or not it is associated with alcohol consumption, plays an important role in the occurrence of EDKA, as documented in several case reports [119,120,121]. This indicates that patients on SGLT2 inhibitors should avoid carbohydrate-free or low-carbohydrate diets and alcohol consumption.

Thus, the carbohydrate deficiency will activate compensatory mechanisms in the part of the body that will be oriented towards balancing this deficiency with the triggering of ketoacidosis under conditions of normal blood glucose values. In this way, the body’s physiopathological reactions will be triggered in a manner similar to those produced in conditions of starvation and dehydration.

### 5.2. CVD

SGLT inhibitors represent a group of substances that, in addition to their blood glucose-lowering effect, have both direct and indirect beneficial effects in the treatment of CVD. This class of drugs can be used in some situations of cardiovascular lesions that accompany DM2 and also in some independent CVDs without a diabetic component.

#### 5.2.1. Diabetic Cardiomyopathy

SGLT inhibitors have been shown to reduce cardiovascular death rates in diabetic patients [122]. Diabetic cardiomyopathy is a systolic and diastolic dysfunction that occurs in the context of DM without ischemic heart disease, hypertension, or valvular lesions [123,124]. Some studies have shown that SGLT2 inhibitors are able to reduce diastolic blood pressure (BP) but without affecting heart rate [125,126,127,128,129], with the most intense effect belonging to Canagliflozin [125]. Empagliflozin significantly reduced both systolic and diastolic BP only in patients with DM2 [130]. In a study that included 75 patients who were administered Canagliflozin 10 mg in a single dose per day, for a period of 24 months, a reduction in systolic and diastolic BP was observed in diabetic patients but not in non-diabetic ones [131]. There is a hypothesis according to which a decrease in systolic and diastolic BP after the administration of SGLT2 inhibitors is due to the sympathetic modulation that they may exert on specific brain areas subject to autonomic control [132,133].

In general, the accumulation of advanced glycated end products plays a key role in the development of diabetic cardiomyopathy. Ectopic accumulation of free fatty acids, alterations in myocardial calcium homeostasis, increased oxidative stress, and mitochondrial dysfunction also have an important contribution [134]. SGLT2 inhibitors are able to partially reverse these modifications and have been shown to improve altered mitochondrial metabolism produced by lipotoxicity associated with the overexpression of peroxisome proliferator-activated receptors-γ [124]. The mechanism was demonstrated using Empagliflozin, which stimulated phospho AMP-activated protein kinase α2, responsible for maintaining the metabolic balance of mitochondrial fatty acid oxidation [123,135]. In a study of 8-, 12- and 16-week-old diabetic db/db mice with diabetic cardiomyopathy, it was observed that the expression of mitochondrial 3-hydroxy-3-methylglutaryl-coenzyme A (mHMG-CoA) synthase is increased while the expression of the enzymes succinyl-Coa-3-oxoacid-CoA transferase 1/3-oxoacid-CoA-transferase 1 (SCOT) and mitochondrial 3-hydroxybutyrate dehydrogenase (mBDH) is decreased. If these mice were treated with Empagliflozin, the activity of mBDH, SCOT and mHMG-CoA synthase was increased, enhancing both the production and utilization of ketone bodies at the cardiac level [136]. These processes are illustrated in Figure 5a,b. In general, the equilibrium of the β-hydroxybutyrate/acetoacetate reaction is controlled by the availability of NADH+H^+^/NAD^+^ [137], with ketosis producing increases of the NAD^+^/NADH+H^+^ ratio in the brain and tissues [138]. Cardiac muscle has an intense metabolism using fatty acids as a preferential energy substrate that β-oxidizes to produce ATP. Other alternative substrates are glucose and ketone bodies. In heart failure, the ability of the myocardium to β-oxidize fatty acids is reduced. Under conditions of satisfactory external nutritional intake, glucose becomes preferentially used under the action of insulin, while in the case of low-carbohydrate diets or starvation, fatty acids and then ketone bodies serve as the main energy source [139,140]. Ketogenic and carbohydrate-restricted diets as well as prolonged fasting have demonstrated their effectiveness in reducing body weight, blood pressure and blood glucose levels [141]. The other side of exogenous ketone bodies is the possibility of long-term alterations in cardiac function [142]. A ketogenic diet and administration of β-hydroxybutyrate reduce the β-oxidation of fatty acids and their uptake by myocytes [143,144]. An increase in myocardial acetyl-CoA with the use of exogenous β-hydroxybutyrate could be responsible for the inhibitory effect on the β-oxidation of fatty acids by modifying the NADH+H^+^/NAD^+^ or acetyl-CoA/CoA-SH ratio [143]. C57BL/6J mice with induced ischemic heart failure that received glucose, palmitate and β-hydroxybutyrate and were fed a ketogenic diet showed a 56% decrease in ejection fraction [142]. High circulating acetoacetate concentrations have been associated with a high number of deaths in patients with heart failure, without being significantly influenced by stress. In addition, they have a harmful potential on vessels, with increased LDL levels, arterial stiffness and functional endothelial damage after prolonged ketogenic diets [145]. The dual nature of ketone bodies (beneficial and harmful) may be time-dependent. Short-term exogenous administration or intermittent ketogenic diets may present an advantage, whereas long-term exposure to elevated ketone body concentrations may have potentially detrimental effects [145,146]. It is known that patients with heart failure and reduced ejection fraction have a high consumption of β-hydroxybutyrate [147]. Starting from the finding of Nielsen et al. [148] regarding the improvement of cardiovascular hemodynamics with an 8% increase in ejection fraction and a 40% increase in cardiac output after exogenous administration of β-hydroxybutyrate [148], the benefit of exogenous administration of ketone bodies versus SGLT2 inhibitors was further investigated [143,144]. Dapgliflozin was found to increase ketone body consumption, increasing myocardial ketolysis by 50%, reducing pyruvate oxidation, increasing the plasma concentration of free fatty acids produced by lipolysis and directing them towards ketogenesis without significantly influencing β-oxidation. A decrease in the NADH+H^+^/NAD^+^ ratio and a reduction in oxidative stress, both materialized by the improvement of ejection fraction after 3 months, were also investigated. In comparison, exogenous β-hydroxybutyrate supplementation accelerates myocardial ketolysis with reduced β-oxidation of fatty acids but without altering glucose uptake and pyruvate oxidation. It also produces a decrease in free fatty acid uptake and an increase in the NADH+H^+^/NAD^+^ ratio [143,144]. Consequently, in the heart, SGLT2 inhibitors cause an increased synthesis of ketone bodies in hepatocyte mitochondria via both major and minor pathways. These ketone bodies will be catabolized in cardiomyocyte mitochondria, a process enhanced by SGLT2 inhibitors [149]. Also, it has been demonstrated that Ertugliflozin promoted the use of ketone bodies as an alternative energy source for cardiomyocytes, along with a reduction in ventricular remodeling, fibrosis, and inflammatory phenomena [122]. Dapagliflozin has shown a suppression of collagen formation with an antifibrotic effect in the heart [150]. Thus, SGLT2 inhibitors contribute to improving myocardial metabolism by reducing preload and myocardial oxidative stress by limiting the phenomena of myocardial fibrosis and necrosis as well as those of arterial stiffness [150]. To these, a decrease in circulating volume due to osmotic diuresis is added, improving the heart workload.

#### 5.2.2. Arrhythmias

In general, SGLT2 inhibitors have been shown to reduce the risk of atrial arrhythmias such as atrial fibrillation and atrial flutter by 24% but have no effect on ventricular arrhythmias [151], while in patients with DM2, SGLT2 inhibitors reduced atrial arrhythmic events by 19% [152]. Furthermore, the actual mechanism by which this phenomenon occurs is still not fully elucidated. However, the disruption of calcium and sodium homeostasis plays an important role in the development of ectopic foci. It seems that an important role played by SGLT2 inhibitors is the handling of Ca^2+^ cations at the cardiomyocyte level. Dapagliflozin and Empagliflozin stimulated the activity of proteins in sarcoplasmic reticulum, restoring Ca^2+^ balance at this level [153]. On the other hand, the excess intracellular sodium provided in cardiomyocytes by sodium–glucose cotransporters can also be an arrhythmogenic point. SGLT inhibitors reduce intracellular sodium overload, thus also having an anti-arrhythmic effect [154]. Wu VC et al. compared the ECG changes of patients treated with SGLT2 inhibitors in DM2 with those of patients who were not treated with these drugs and concluded that there was no difference between the two groups [155]. A hypothesis regarding these aspects would be that, over time, SGLT inhibitors, by reducing necrosis and fibrosis phenomena, may have an important role in stopping the evolution of cardiac lesions towards atrial arrhythmias, since the fibrotic/necrotic myocardial tissue itself could be an arrhythmogenic source.

#### 5.2.3. Coronary Artery Disease

The main points where SGLT inhibitors can intervene in ischemic heart disease are as follows: improving the lipid profile and atherosclerotic phenomena, reducing ventricular remodeling, and treating heart failure after acute myocardial infarction in patients with and without DM2.

The development of atherosclerotic phenomena can be influenced by SGLT2 inhibitors both through a direct effect on atheromatous plaque and by influencing the lipid profile. To improve endothelial dysfunction, SGLT2 inhibitors act both on endothelial cells and on macrophages at this level, preventing the accumulation of oxidized LDL. This accumulation will cause a release of proinflammatory cytokines belonging to the interleukin (IL) class, as well as intercellular cell adhesion molecule-1 (ICAM-1) and vascular cell adhesion protein-1 (VCAM-1), and also the stimulation of monocytes’ migration to the endothelium and their activation to macrophages through macrophage colony-stimulating factor. In this regard, Dapagliflozin, Empagliflozin, Canagliflozin and Luseogliflozin showed a decrease in the expression of not only monocyte chemoattractant protein-1 (MCP-1), VCAM-1 and ICAM-1 but also IL-1β, IL-6 and IL-18 [156,157,158]. Another important factor in the development of endothelial dysfunction is a reduction in available NO under the action of hyperglycemia and oxidative stress that directly reduces the activity of NO synthase (NOS). An increase in NOS activity is produced by Empagliflozin and Canagliflozin [156,157,158], while Dapagliflozin increases the availability of NO both through a direct effect influenced by the ethyl ether group [58] and by ameliorating oxidative stress (tumor necrosis factor-α (TNF-α)-induced) [156,157,158]. Another interesting aspect in the pathophysiology of coronary heart disease, but also in other heart diseases, is the effect of SGLT2 inhibitors on lipid profiles. In general, it is observed that SGLT2 inhibitors reduce plasma triglyceride (TGL) concentration but with an increase in LDL and HDL without affecting the LDL/HDL ratio [130,159]. In a study [131], a decrease in total cholesterol concentration was observed after Dapagliflozin administration in patients with CKD, regardless of DM stage, but without significant changes in TGLs, LDL and HDL. There were only significant declines in LDL for patients with DM and high BP, not for those without DM and high BP [131]. One hypothesis can be caused by an increase in the expression of HMG-CoA reductase, a reduction in LDL receptor expression and a decrease in the hepatic uptake of the LDL fraction, thus obtaining a decrease in the circulating LDL clearance [130,160]. SGLT2 inhibitors also act by reducing the activity of the enzymes TGL-lipase and lipoprotein-lipase. This inhibition would actually be an indirect effect because, in the context of SGLT2 inhibitor medication, a decrease in the expression of angiopoietin-like protein, which in turn inhibits lipoprotein-lipase, was observed [159,161]. In contrast, Lund S.S et al. [162] attribute these increases in LDL and HDL to hemoconcentration. This effect is not observed in the case of increased TGLs, which is not fully explained by hemoconcentration theory in this case. Most likely, all these hypotheses added together bring an argument for the global effect observed in the lipid profile.

Mitophagy, an important process in maintaining myocardial cell function, can be activated in a ubiquitin-dependent or -independent manner. The most important pathway mediated by the activation of the kinase-ligase enzyme system is the PTEN-induced kinase 1 (PINK1)–Parkin pathway [163,164]. On the other hand, this process must be in balance with mitochondrial biogenesis to ensure efficient myocardial function. SGLT2 inhibitors are able to promote cardiomyocyte mitochondrial integrity by regulating both mitophagy and mitochondrial biogenesis [165]. Yang et al. [164] demonstrate the involvement of Canagliflozin in both mitophagy and biogenesis. Mitophagy is promoted by the activating action of Canagliflozin on the PINK1–Parkin pathway. In intact mitochondria, PINK1 is transported to the inner mitochondrial membrane and protease-degraded [163,166]. A decrease in membrane potential will block PINK1 transport, producing dimerization and autophosphorylation (these two processes activate PINK1) [164,167]. In this form, PINK1 recruits Parkin and triggers the process of mitophagy, a mechanism activated by Canagliflozin in C57BL/6J mice with streptozocin-induced DM and fed with a high-fat diet (HFD) [164]. In the same mice, Canagliflozin improved mitochondrial biogenesis by upregulating the peroxisome proliferator-activated receptor G coactivator 1-α (PGC-1α)–mitochondrial transcription factor A (TFAM) pathway, a pathway with reduced expression under hyperglycemic conditions [164]. On the other hand, sirtuin-1 (SIRT-1) activated by SGLT2 inhibitors can directly induce autophagy by deacetylating autophagy-related genes [168]. Empagliflozin upregulates AMP-activating protein kinase (AMPK) by promoting its phosphorylation to the form in which it activates PGC-1α-TFAM [135,169,170,171]. Empagliflozin can reduce, on one hand, the excessive upregulation of dynamin-related protein 1 (Drp1) involved in mitochondrial fission and downregulated in diabetic Sprague–Dawley rats [170]; on the other hand, it attenuates the depletion of mitochondrial fusion proteins mitofusin 1 (Mfn1), Mfn2 and optic atrophy 1 (OPA1), proteins that are downregulated in mitochondrial alterations, thus lowering these modifications [170,171]. Empagliflozin can reverse myocardial microvascular damage resulting from DM by inhibiting mitochondrial fission mediated by AMPK [170]. Similar phenomena determined for Dapagliflozin have been observed in skeletal muscle tissue in Sprague–Dawley rats with induced DM2 [172]. In the late post-myocardial infarction period, SGLT2 inhibitors have demonstrated efficacy when included in long-term therapeutic regimens of diabetic and non-diabetic patients. In this context, they are useful for reducing ventricular remodeling and improving heart failure. Empagliflozin has been shown to reduce left ventricular end-systolic and end-diastolic volumes [173]. This effect is a consequence of the protective action of mitochondrially induced autophagy via the Beclin-1-dependent pathway, with upregulation of Beclin-1 being responsible for triggering autophagy during reperfusion [173,174]. Empagliflozin enhances the binding of toll-like receptor 9 (TLR9) to Beclin-1, increasing TLR9 activation in mitochondria [174,175,176]. Under this mechanism, Empagliflozin increases the binding of Beclin-1 and NAD-dependent deacetylase SIRT-3 [174], with the Beclin-1-dependent pathway being an important process in the induction of autophagy under ischemia–reperfusion conditions [175]. Here, the activated Beclin-1–TLR9–SIRT-3 complex increases mitochondrial respiration and improves cardiomyocyte protection against reactive oxygen species (ROS) and apoptosis [177].

However, in the presence of coronary artery disease and localized or generalized atheromatosis, a question arises regarding the impact of hemoconcentration’s secondary effect related to the use of SGLT2 inhibitors in producing osmotic diuresis. Usually, hemoconcentration can lead to an increase in the risk of thrombosis and lower limb amputations below the knee, as observed in patients with DM2 [178]. The reduction in systolic and diastolic blood pressure is another factor responsible for the occurrence of peripheral arterial disease and lower limb amputation, favoring local ischemic conditions [179,180]. This risk is higher in patients with DM and a history of peripheral ischemia, ulcers and infections of the lower limbs [181,182]. In a meta-analysis study, the results of 31 randomized controlled trials dedicated to peripheral artery disease and 15 to amputations were investigated. Here, slightly increased risks for amputations and peripheral artery disease were identified among subjects who used SGLT2 inhibitors compared with those who received antidiabetic drugs from other drug classes. An increased risk was observed only with the use of Canagliflozin [178]. A data analysis including the trials EMPA-KIDNEY (containing patients with DM2+CDK), EMPA-REG OUTCOME (containing patients with DM2+/−CVD) and EMPEROR (containing patients with heart failure +/−DM2) showed that Empagliflozin does not present an increased risk of amputations, bone fractures and Fournier’s gangrene (FG) [183]. On the other hand, the Canagliflozin Cardiovascular Assessment Study (CANVAS) reveals a two-fold increase in the risk of lower limb amputations [181,182]. In a systematic review, Nani A et al. [182] performed a meta-analysis that included 42 randomized trials. Here, amputations were described in 34 trials, while lower limb fractures were evaluated in 33 trials. The highest risk was observed when Canagliflozin was administered, followed by Dapagliflozin and Ertugliflozin, whereas Empagliflozin and Tofogliflozin were not associated with amputations. In contrast, all SGLT2 inhibitors presented a risk of bone fractures [182]. Other meta-analyses that evaluated the risk of lower limb amputation and bone fractures did not associate the incidence of these events with SGLT2 inhibitors, the values being close to those of the control group [184], while others associate gliflozins with an increased frequency of both bone fractures and lower limb amputations [181,185]. It seems that SGLT2 inhibitors demonstrated a protective effect in large vessels, whereas the effect was opposite in small vessels because of pre-existing lesions in DM2 associated with phenomena that promote stasis such as reduced blood pressure and hemoconcentration. SGLT2 inhibitors may have an increased risk of lower limb amputation, drug-related and/or patient-related.

### 5.3. Non-Alcoholic Fatty Liver Disease (NAFLD)

Different therapeutic effects of SGLT2 inhibitors were also investigated in other diseases. In patients with or without DM, they could be influenced by the reduction in blood glucose levels, the improvement of the lipid profile and the additional anti-inflammatory potential. SGLT2 inhibitors also have beneficial effects in liver diseases characterized by steatosis, especially those occurring in metabolic syndrome.

The first visible effect is a reduction in blood glucose that exposes the hepatocytes to lower amounts of glucose with an increase in the production and release of glucagon, which causes the activation of hormone-dependent lipase in adipose tissue [186]. This activation of the key enzyme in the hydrolysis of TGLs to free fatty acids and glycerol will lead to a reduction in the size of adipose tissue. Among the drugs used in the joint management of DM2 and NAFLD, including refractory forms and those progressing to liver cirrhosis, gliflozins, especially, Empagliflozin, Canagliflozin and Ipragliflozin, have shown a high potential [187]. In addition to the benefits mentioned above, SGLT2 inhibitors also contribute to the reduction in hepatic fat accumulation and inflammation, and in the case of cirrhosis, they contribute by increasing sodium and water excretion [188]. A meta-analysis of SGLT2 inhibitors in NAFLD evidenced a significant reduction in aspartate aminotransferase (AST), alanine aminotransferase (ALT), liver fibrosis and visceral fat [189]. SGLT2 inhibitors appear to promote fatty acid β-oxidation in the liver and fatty acid transport to the liver. In a study conducted on ApoE(−/−) knockout mice with HFD, when Empagliflozin was administered, it was found that total cholesterol and TGL values were reduced, along with reduced expression levels of lipogenic enzymes such as fatty acid synthase (FAS), sterol regulatory element-binding protein-1 (SREBP-1), and phosphoenolpyruvate carboxykinase-1 (PEPCK-1) compared to control groups, potentially alleviating NAFLD [190]. Similar results in HFD murine models have been observed for Canagliflozin [191]. In fact, SREBP-1 activation is one of the regulatory factors of lipogenesis and has a modulatory effect on FAS, acetyl-CoA carboxylase 1 (ACC1) and sterol-CoA desaturase 1 (SCD1), with an increase in SREBP-1 expression being associated with liver deterioration [192,193] due to the fact that SREBP-1 promotes hepatic synthesis of fatty acids and TGLs [193]. Downregulation of SREBP-1 expression by SGLT2 inhibitors has been observed in several studies on laboratory animals, as reviewed by Khaznadar F et al. [194].

It has been observed that gliflozins reduce insulin resistance by decreasing hepatic diacylglycerols, compounds associated with increased insulin resistance, and downregulate cellular signaling through the mammalian target of rapamycin (mTOR) pathway [195]. Also, a reduction in proinflammatory cytokines such as TNF-α, MCP-1, IL-1β, and IL-6 has been observed under gliflozin treatment, ameliorating NAFLD in experimental murine models [190,191]. The release of these proinflammatory cytokines is directly regulated in the mTOR pathway [196]. SREBP-1 is also directly influenced by mTOR; gliflozins reduced mTOR expression, leading to a decrease in SREBP-1 activity [192].

In this way, SGLT2 inhibitors demonstrate their efficacy in improving NAFLD by inhibiting upstream cellular signaling pathways involved in lipid accumulation in the liver and downstream cellular signaling pathways, influencing the activity of enzymes with a lipolytic effect.

### 5.4. Renal Diseases

In general, all patients with DM2 have more or less impaired renal function. However, the effects of SGLT2 inhibitors are not limited to patients with DM but extend to other categories of patients without DM. The effects of SGLT2 inhibitors are reflected in preventing a decline in the estimated glomerular filtration rate (eGFR) of patients with CKD and end-stage kidney disease (ESKD) [197,198]. Evidence from recent clinical trials reinforces the established benefits of SGLT2 in patients with heart failure and CKD management. A multicenter, randomized trial shows that after 12 weeks of treatment with dapagliflozin, the symptoms and physical status of patients with heart failure and preserved ejection fraction improved considerably [199]. According to data obtained from a study involving 507 critically ill patients, even though organ dysfunction was not significantly improved by the use of dapagliflozin, its efficiency was evidenced by good toleration in this high-risk category [200]. A reference clinical study involving the therapeutic effects of SGLT2 is EMPA-KIDNEY, which confirmed the cardiorenal protective action of SGLT2 inhibitors, supporting their use in both the primary prevention and treatment of CKD [201]. Findings from the DECLARE-TIMI 58 trial showed that Dapagliflozin decreases the eGFR and albuminuria and has a key role in the prevention of diabetic kidney disease [197,202].

Some mechanisms by which gliflozins improve renal function are specific to the kidney; others, such as the inhibition of the release of proinflammatory molecules, are observed in several organs and systems.

Like all substances with a diuretic effect, gliflozins have several advantages over classic diuretics: they do not cause major electrolyte imbalances, they are not associated with hyperkalemia, they reduce the risk of acute kidney injury and they do not produce prominent neurohormonal stimulation [203]. In general, to obtain the maximum benefit from these drugs, an eGFR of at least 30 mL/min is needed, but new guidelines recommend that the eGFR be greater than 20 mL/min/1.73 m^3^ [204]. The increase in sodium concentration in the renal tubular lumen as a result of SGLT2 inhibition will cause an osmotic gradient that exceeds the capacity of Na^+^/K^+^ ATPase in the macula densa. This will cause the hydrolysis of ATP to adenosine under the action of an extracellular nucleotidase. Adenosine will attach to adenosine A1 receptors, with vasoconstriction of the afferent arteriole. Vasoconstriction will cause a reduction in glomerular blood flow with a decrease in intraglomerular pressure. As a consequence of the effects of adenosine on the juxtaglomerular apparatus, there will be a decrease in renin release and a decrease in vasoconstriction mediated by RAAS [197,205]. This removes the effects of angiotensin II, which has a positive impact, improving both the eGFR and heart function in CVD by lowering blood pressure values and reducing cardiac workload. However, a study involving eight-week-old male Dahl SS rats did not identify any effect of Dapagliflozin on the intrarenal RAAS in conditions of salt-induced hypertension [206]. As a conclusion reflected here, this mechanism is still incompletely elucidated, requiring further studies to clarify this aspect. Human renal proximal tubular epithelial cells (HPTCs) were incubated with Dapagliflozin and Empagliflozin in an experimental diabetic environment where both gliflozins were able to inhibit ICAM-1 expression [207]. In another study performed on HPTC, Empagliflozin caused a reduction in basal and endothelin-1-stimulated IL-1β expression, while the induced expression of MCP-1/CC-chemokine ligand-2 and IL-6 was decreased by Canagliflozin [208]. However, it is known that production of proinflammatory cytokines such as IL-1β and TNF-α in the kidney [207,209] suppresses the production of erythropoietin (EPO) [210]. By suppressing these molecules, gliflozins could have a beneficial effect on the increased release of EPO in CKD. On the other hand, when hypoxia is present, EPO production is stimulated by the release of both the HIF-1α and HIF-2α isoforms of hypoxia-inducible factor (HIF). SGLT2 inhibitors appear to selectively stimulate HIF-2α [211,212] and also enhance the activity of SIRT-1, a direct activator of HIF-2α [211]. Some studies have shown that SGLT2 inhibitors are able to reduce ferritin, transferrin desaturation and hepcidin activity by improving iron absorption from the digestive tract and the mobilization of iron from tissue stores and macrophages [212,213]. The advantage would be that SGLT2 inhibitors enable hemoglobin levels to rise without the need for iron supplements [214], an advantage also observed in patients with cardiorenal syndrome [215]. Docherrty K.F et al. [216] studied this effect in the DAPA-HF trial, where the administration of Dapagliflozin for 12 months in patients with heart failure NYHA classes II-IV and iron deficiency induced changes in the mean values of hepcidin from 24.3 ng/mL to 17.2 ng/mL and ferritin from 158.2 ng/mL to 130 ng/mL. Similar results were observed with Canagliflozin in CREDENCE trial, which included diabetic patients with CKD [217], as well as in a post hoc analysis of the DELIGHT trial with similar cohorts of patients where Dapagliflozin, along as earlier results, also produced an increase in erythropoietin to a mean of 82.9 pg/mL, compared to placebo (79.8 pg/mL) [218]. Murashima M et al. [219] demonstrated that the administration of SGLT2 inhibitors improved anemia in diabetic patients with an eGFR of 15–30 mL/min/1.73 m^2^. Due to the low production of EPO, the increase was smaller in patients with eGFRs ≤ 15 mL/min/1.73 m^2^ than in those with eGFRs above 60 mL/min/1.73 m^2^. According to the IRONMAN trial, a significantly higher increase in hemoglobin was observed when the intravenous administration of an SGLT2 inhibitor was combined with ferric derisomaltose [220]. In this way, gliflozins help to treat anemia in CKD directly by removing EPO-inhibiting factors (proinflammatory substances) and indirectly by stimulating EPO production through HIF-2α and SIRT-1, as well as their impact on iron store mobilization. Figure 6 displays mechanisms involving SGLT2 inhibitors in improving anemia.

One of the adverse reactions in the urinary tract is that SGLT2 inhibitors favor both bacterial and fungal colonization due to the high glucose content in urine [221]. Glycosuria induces the development and proliferation of Candida species, especially Candida albicans, leading to candidiasis sepsis [222], as well as favoring *E. coli* infection [221]. A study revealed an increased association of SGLT2 inhibitors with candidiasis infections and no correlation with bacterial infections [223]. Even if the incidence of these infections is increased, as other authors have reported [224,225], the benefits outweigh these shortcomings in most cases; therefore, the discontinuation of gliflozin medication under conditions involving urinary tract infection is not recommended [224]. Another concern is dedicated to Fournier’s gangrene (FG), an infectious necrotizing fasciitis of the perineum and genitals in a fulminant form, found associated with SGLT2 inhibitors in the context of glycosuria [226,227]. Although no exact explanation for the occurrence of FG was found [227], there are studies where no correlation between FG and gliflozins was demonstrated [228]. According to the FDA and the Medicines and Healthcare Products Regulatory Agency, special attention should be paid to FG in gliflozin users [228]. In a data analysis that involved 78 patients hospitalized for FG who required surgical reconstruction, of which 41% had DM, none of them were treated with SGLT2 inhibitors [228]. Despite these findings, cases of FG associated with SGLT2 inhibitors have been reported in the literature. A literature review that included data from 12 patients diagnosed with FG highlighted 5 cases determined by Empagliflozin, 5 patients by Dapagliflozin and 2 patients by Canagliflozin [226]. Another review identified 13 case reports of FG; 8 were attributable to Empagliflozin, while the rest were attributable to both Empagliflozin and Canagliflozin [229]. Additional new cases have been reported in the last year [230,231]. In the event of FG, treatment with an SGLT2 inhibitor should be stopped and not reinstated after recovery [226]. Considering these aspects, even though FG is a rare event, it should not be ignored in patients treated with SGLT2 inhibitors.

### 5.5. Obesity

The production of a caloric deficit in overweight and obese people can be achieved by using SGLT inhibitors. Due to the elimination of glucose from the body, calories are lost, and a depletion of 60–100 g glucose/day [232] is equivalent to 300 kcal/day [233]. SGLT1 inhibition at the intestinal level achieves a synergistic effect in this regard because the absorption of intestinal glucose in the body is reduced by exogenous intake. On the other hand, glycosuria and the inhibition of intestinal absorption induce the mobilization of lipid stores and the use of fatty acids as an energy source, determined by their entry into β-oxidation or their transformation into ketone bodies, a situation similar to that encountered in prolonged fasting. Empagliflozin and Canagliflozin have been shown to stimulate lipolysis and reduce lipogenesis and lipid accumulation in the liver, along with lipid store mobilization in subcutaneous adipose tissue [234]. Among the hypotheses that would provide additional explanations for the role of gliflozins in weight loss is the link between obesity and inflammation. The inhibition of the mTOR pathway by SGLT2 inhibitors could lead to fasting-like mitochondrial changes [233]. Experimentally, it was observed that 6-week-old male C57Bl/6 mice fed with a high-fat diet and treated with Empagliflozin exhibited protection against weight gain, insulin resistance and hepatic steatosis in comparison to control subjects [235]. Empagliflozin leads lipid metabolism towards lipolysis and the use of lipids as an energy resource by increasing the activity of AMPK with ACC1 phosphorylation in skeletal muscle [130]. Moreover, Empagliflozin reduces hepatic lipogenesis and gluconeogenesis by increasing the hepatic expression of both SIRT-1 and AMPK [236], with a direct effect on reducing the activity of PEPCK-1, a key enzyme in gluconeogenesis [190]. Empagliflozin also affects pyruvate dehydrogenase kinase-4 (PDK-4), an enzyme that regulates the activity of pyruvate dehydrogenase complex by its inactivation through phosphorylation. The decrease in PDK-4 expression after Empagliflozin administration was accompanied by a reduction in fatty acid uptake in the liver, suggesting that the protective effect against obesity would actually be the alteration of PDK-4 induced by Empagliflozin [235]. In conclusion, in addition to the major role of reducing glucose calories at the renal level, the activation of signaling pathways and the influence of key enzymes recommend SGLT2 inhibitors as beneficial in metabolic syndrome. Here, the alteration of lipid and carbohydrate metabolism in the sense of adipose tissue accumulation accompanies and favors damage to vital organs: the heart, liver and kidneys.

As recently demonstrated [237], adipose tissue has an epigenetic memory of obesity; many differentially expressed genes in adipocytes, adipose progenitor cells, lipid-associated macrophages, perivascular macrophages, non-perivascular macrophages, endothelial cells and mesothelial cells remain altered even after weight loss, thus justifying one of the processes responsible for long-term failures in weight loss attempts. At the same time, the behavior of adipose stem cells and mesenchymal cells can be modified; they exhibit varying behaviors contingent upon the conditions to which they are exposed in vitro [238,239]. This last finding may bring hope in the search for agents capable of permanently modifying the behavior of cells in adipose tissue, and in this direction, SGLT inhibitors could have a role that needs to be investigated in further research.

## 6. Off-Label Uses of SGLTs with Promising Therapeutic Potential and Future Directions

Based on the mechanisms and pharmacological effects of SGLT inhibitors discussed above, there are some studies and trends related to their off-label use. Some uses are indirect findings and extrapolations of known pharmacodynamic actions, but others are new research and directions.

### 6.1. Anti-Aging and Age-Related Processes

The effect of SGLT inhibitors is attributed to the revitalization of some functions of vital organs, with an improvement in their status being a gain from the point of view of increasing the quality of life and prolonging life. At the mitochondrial level, this can be achieved by activating pathways involved in longevity, such as AMPK, SIRT-1 and SIRT-3, by compensating for mitochondrial deficits and prolonging the lifespan of mitochondria or mitochondrial fusion, as well as improving the control of mitochondrial function and mitophagy or mitochondrial fission [240]. Another effect through which SGLT2 inhibitors could exert an anti-aging effect is the reduction in inflammatory phenomena exerted by an excess of free fatty acids that attach to TLR2 and TLR4, especially in skeletal muscle, the liver and adipose tissue. The consequence of this attachment is an increase in insulin resistance, which is also promoted by activation of nuclear factor kappa B (NF-kB) [241]. The downregulation of NF-kB may additionally be produced by the activation of both SIRT-1 and AMPK by SGLT2 inhibitors [242]. Autophagy can also be improved by modulating AMPK, mTOR, SIRT-1 and HIF, the main regulators of this process [241,242,243]. The stimulation of ketogenesis and the enhancement in free fatty acid transport to the liver by gliflozins contribute to the removal of free fatty acids and stop the phenomena triggered by their excess [244]. As a result, the overall effects, encompassing the reduction in plasma glucose and free fatty acid concentrations, reduction in inflammatory phenomena and modulation of metabolic and cellular signaling pathways, contribute both directly and indirectly to a general anti-aging effect.

### 6.2. Allergic Bronchial Asthma

The incidence of bronchial asthma (BA), chronic obstructive pulmonary disease (COPD) and other respiratory diseases was diminished in patients receiving SGLT2 inhibitors across various treatment regimens [245]. The exact mechanism by which gliflozins improve bronchial hyper-reactivity is still unknown, but one of the factors seems to be the effect of preventing/reducing pulmonary fibrosis [246]. In BALB/c mice, Lee Y.E and Im D.S [247] demonstrated an inhibitory effect on mast cell degranulation after Empagliflozin and Canagliflozin administration. A suppressive effect of the proinflammatory cytokines IL-4, IL-13, IL-17 and interferon γ (INF-γ) involved in the initiation of allergic BA was also observed [247]. Among all SGLT2 inhibitors investigated, Dapagliflozin seems to be most effective in preventing BA attacks [248]. However, other authors have not noted differences in the symptoms and incidence of BA exacerbations in patients receiving SGLT2 inhibitors in comparison to those receiving metformin [249]. One hypothesis for these findings is that the observations were made in a Japanese population, where certain isoenzymes or particular signaling pathways may show small variations expressed by a different response to off-label data. It would be a challenge to explore these small differences because big therapeutic responses could be found between them.

### 6.3. Other Contributions Worth Considering for Future Use: Rare Forms of Neutropenia

The neutropenia characterized by the accumulation of 1,5-AG-6 phosphate found in glycogen storage disease type 1, congenital neutropenia type 4 (associated with glucose-6-phosphate transporter deficiency and glucose-6-phosphatase catalytic subunit-3 deficiency), may respond to SGLT2 inhibitors [44,46]. This neutropenia occurs due to the lack of the corresponding hexokinase responsible for the hydrolysis of phosphate bonds. Gliflozins, which cause an increase in the glucose level in renal tubules for excretion, produce an indirect inhibition of SGLT5 due to the fact that a large amount of glucose at this level competitively inhibits SGLT5 [46]. The outcome is the inhibition of 1,5-AG reabsorption at the renal tubular level with the promotion of its elimination, and thus, the plasma concentration of 1,5-AG decreases, and the use of gliflozins results in the amelioration of neutropenia [250].

Occasionally, novel applications emerge from mechanisms that are known and described for applications already documented, sometimes disclosing further pathways that may be helpful in some rare diseases. Observations on SGLT2 inhibitors in other clinical applications may lead to discovery of new mechanisms based on what is already known as well as empirical observations, opening therapeutic doors for new future research.

## 7. Conclusions

The chemical structure of SGLT inhibitors is an essential factor influencing the affinity and selectivity for SGLT1 versus SGLT2. Modifying glucose residues prolongs the inhibition of SGLT2 due to the structural similarity. Also, changing the C-O glycosidic bond to a C-C bond improves the molecules’ pharmacokinetics by making them resistant to digestive degradation. With the introduction of gliflozins in therapeutics as antidiabetic agents, new properties have been highlighted, actively contributing to the improvement of changes encountered in metabolic syndrome in apparatuses and systems. In addition to enhancing plasma glucose levels and lipid profiles, important factors in the evolution and development of CVD, NAFLD and CKD have induced mitochondrial changes that can improve vascular lesions and heart, kidney and liver functions. The activation of the inhibition of certain enzymes and cell signaling pathways produced a global effect of lipolysis and the mobilization of fatty acids from stores, with their diversion towards β-oxidation and ketone body synthesis, along with an increase in their utilization. Another important role is the reduction in fibrosis and anti-inflammatory phenomena. Through the caloric deficit created and by facilitating lipid mobilization in tissues, gliflozins enhance obesity management. All these effects contribute to an improvement in patients’ quality of life and physical condition, thereby also demonstrating anti-aging properties. These findings open the possibility of their use in other pathologies where numerous studies exist, sometimes showing controversial results and occasionally demonstrating therapeutic efficiency. All these benefits must be weighed against various detrimental effects, especially EDKA and the risk of lower limb amputations. Minimal structural modifications indicate important changes from a pharmacokinetic and pharmacodynamic point of view. These structures could represent a starting point for modeling both the carbohydrate part and the aglycone to increase the therapeutic spectrum in anti-aging therapies with the reversibility of structural alterations in the physiological aging process. This could also be able to reduce or cancel the rather unpleasant adverse effects of this class.

## Figures and Tables

**Figure 1 ijms-26-06937-f001:**
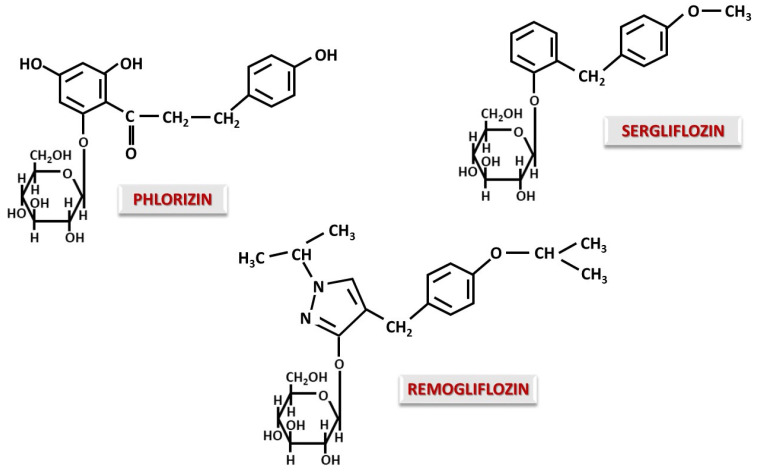
Chemical aspect of Phlorizin, the starting molecule of this class, and SGLT inhibitors with O-glucosidic structure.

**Figure 2 ijms-26-06937-f002:**
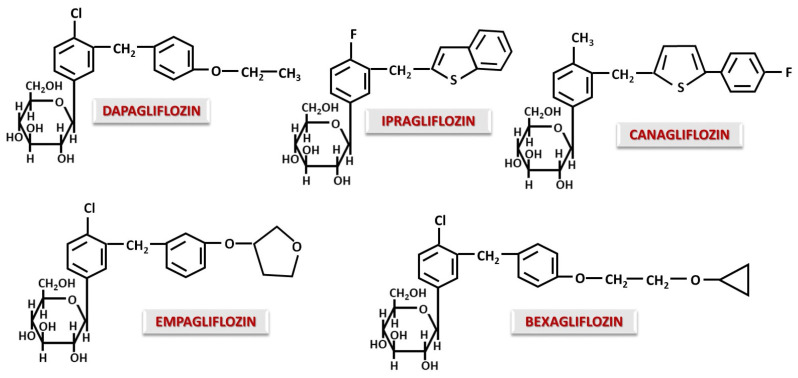
Chemical aspect of SGLT inhibitors with C-glucosidic structure containing compounds with aglycone modifications.

**Figure 3 ijms-26-06937-f003:**
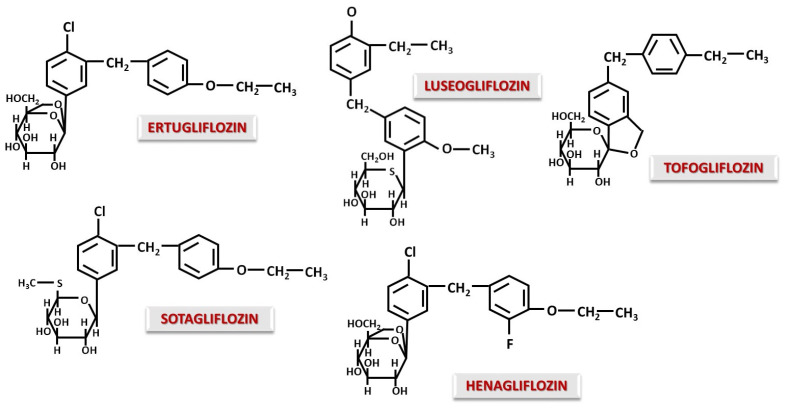
Chemical aspect of SGLT inhibitors with C-glucosidic structure containing compounds with modifications of the carbohydrate moiety.

**Figure 4 ijms-26-06937-f004:**
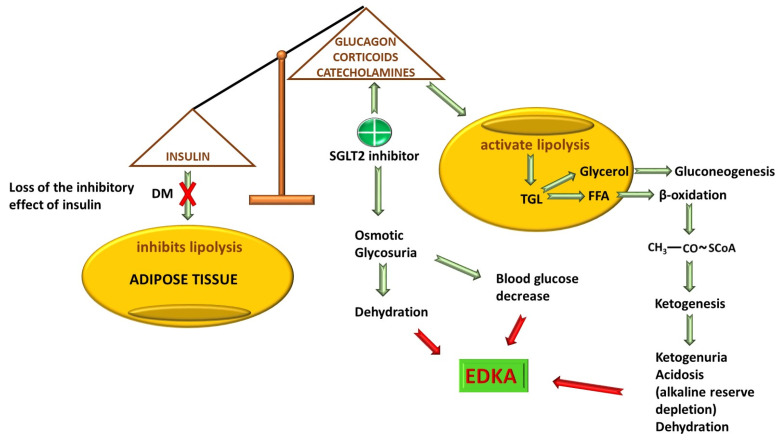
Schematic representation of the pathophysiological processes involved in the development of EDKA. DM—diabetes mellitus; TGLs—triglycerides; FFAs—free fatty acids; EDKA—euglycemic diabetic ketoacidosis.

**Figure 5 ijms-26-06937-f005:**
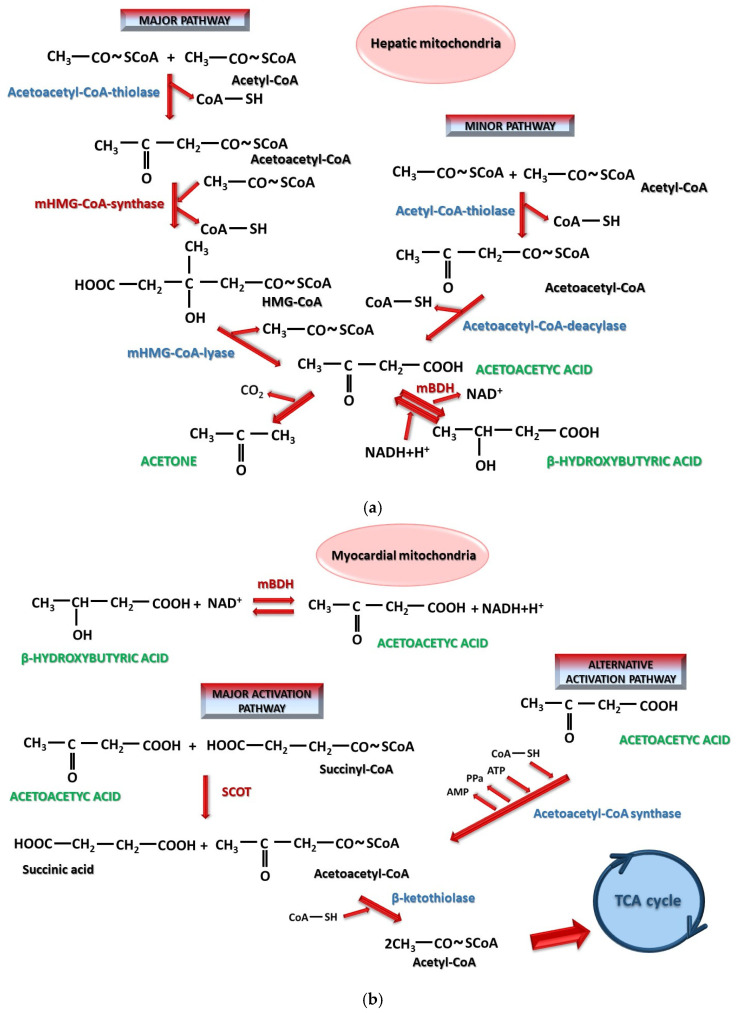
(**a**) Mitochondrial hepatic synthesis of ketone bodies via major and minor pathways, where the enzymes HMG-CoA synthase and BDH, influenced by SGLT2, are marked in red. HMG—3-hydroxy-3-methylglutaryl; mHMG—mitochondrial 3-hydroxy-3-methylglutaryl; CoA—coenzyme A; mBDH—mitochondrial 3-hydroxybutyrate dehydrogenase. (**b**) Mitochondrial myocardial catabolism of ketone bodies with the major and alternative activation pathways of acetoacetic acid, where the enzymes SCOT and BDH, influenced by SGLT2, are marked in red. SCOT—succinyl-Coa-3-oxoacid-CoA transferase 1/1/3-oxoacid-CoA-transferase 1; CoA—coenzyme A; mBDH—mitochondrial 3-hydroxybutyrate dehydrogenase, TCA—tricarboxylic acid.

**Figure 6 ijms-26-06937-f006:**
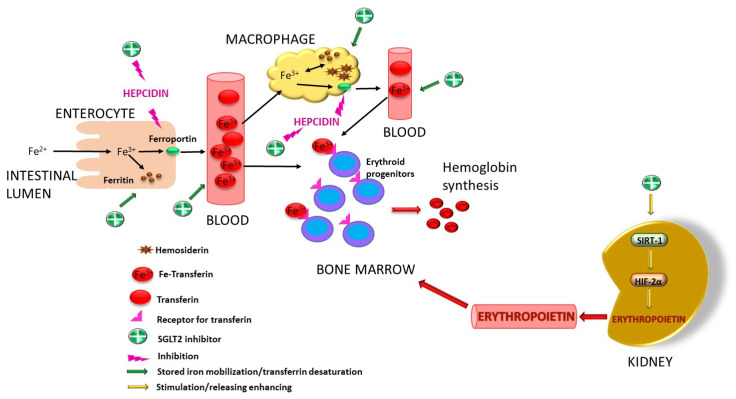
Schematic representation of mechanisms involving SGLT2 inhibitors in alleviating anemia. HIF-2α—hypoxia-inducible factor-2α; SIRT-1—sirtuin-1.

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
