# Peer review of "SGLT2 Inhibitors: From Structure–Effect Relationship to Pharmacological Response"

_ijms, 2025, doi:10.3390/ijms26146937_

Round 1
Reviewer 1 Report
Comments and Suggestions for Authors
The manuscript serves as a detailed overview of SGLT inhibitors, focussing on SGLT2 inhibitors and the mechanisms by which they achieve their therapeutic effects in diabetes mellitus, and other effects beyond DM management. It includes their mechanisms in conditions belonging to the metabolic syndrome and beyond. The review outlines the drugs, the structure-effect relationship of each gliflozin, their beneficial effects in various CVDs; as well as future prospects of this class of drugs, that includes potential application in obesity management and for their anti-ageing effects.
The introduction is clear and comprehensive.
The article provides structured information on the drugs, which will serve as a good reference(quick, specific, up- to- date) for the scientific community, on the beneficial therapeutic effects and adverse reactions.
Strength: While the focus is on SGLT2 inhibitors, there is also much detail on SGLTs and SGLT inhibitors in general.
Weakness: Commonly, the methodology of the review should be included. This is normally done in the form of a table or text outlining how the search was performed, ie keywords used and search engines; and then how many papers were yielded, how many were excluded on what basis, and how many ended up being used in the review. Further it can be indicated how many were found/used per topic of discussion. It would also show the period of publications accepted for the study eg only articles from 2010 were used.
I would strongly recommend that the authors consider including this information. It serves to indicate what the content of the review is, so that the reader can quickly see topics of interest or relevance before reading the whole article.

Author Response
Comment:
Commonly, the methodology of the review should be included. This is normally done in the form of a table or text outlining how the search was performed, ie keywords used and search engines; and then how many papers were yielded, how many were excluded on what basis, and how many ended up being used in the review. Further it can be indicated how many were found/used per topic of discussion. It would also show the period of publications accepted for the study eg only articles from 2010 were used.
I would strongly recommend that the authors consider including this information. It serves to indicate what the content of the review is, so that the reader can quickly see topics of interest or relevance before reading the whole article.
Response:
We introduced „Methodology” section as referee recomended:
„The literature search strategy was carried out in the PubMed, Google Scholar, ScienceDi-rect, Web of Science databases, giving priority to the newest articles in the field. The time interval was generally extended to 10 years, sometimes, where we considered relevant, ar-ticles older by 1 or two years compared to the proposed period were also taken into con-sideration (for example, those with historical value referring to certain discoveries). Meet-ing abstracts and Proceedings, Letters to the Editor, Commentaries, Editorials were ex-cluded, only peer-reviewed articles were taken into account. The main keywords were "SGLT2 inhibitors", "SGLT1", "SGLT2", "SGLT3", "SGLT4, "SGLT5", "gliflozins", "O-glucoside gliflozins", "C-glucoside gliflozins" and combinations between them. Com-binations between "SGLT2 inhibitors" and "euglycemic diabetic ketoacidosis", "diabetic cardiomyopathy", "arrhythmias", "coronary artery disease", "non-alcoholic fatty liver dis-ease", "renal disease", "obesity" were also followed. Last but not least, we used multiple combinations of the keywords listed above. Sometimes, searching for certain information, we found new directions, ultimately resulting in the chapter dedicated to the off-label use of gliflozins. Being drugs with molecular modeling potential open to new current re-search, we emphasized the known relationships between structure and effect as well as the main mechanisms by which SGLT2 inhibitors exert their therapeutic benefits but also adverse reactions, because minor structural changes can sometimes produce major trans-formations.”
Reviewer 2 Report
Comments and Suggestions for Authors
It should be emphasized that the acceptable level of text overlap of 15% confirmed via iThentic for the review paper.
It should be stated that some effects of SGLT2i are still in the realm of assumptions and not claims!
- The introduction should refer to the KDIGO and ADA recommendations for SGLT2i. New pillars of therapy (finerenone) should also be added.
The main focus should be shifted from diabetes to chronic kidney disease, which affects almost every 11 inhabitants of the planet and which accounts for the majority of medically treated CKD patients!
- The importance of therapy in anemia is missing!
- Supplement with effects on lipid profile and diastolic blood pressure!
- Enough redundant text. Some parts should be converted into tables and visual representations. It has potential to be improved. I would suggest a revision to the mayor.
Author Response
Comment 1: - The introduction should refer to the KDIGO and ADA recommendations for SGLT2i. New pillars of therapy (finerenone) should also be added.
Response: „Recent KDIGO guidelines [16,17] have highlighted SGLT2 inhibitors as the first therapeutic option for patients with DM2 and chronical kidney disease (CKD), alongside metformin, RAAS blockers and statins [18] having the concern related to the fact that one in three people with DM also has CKD [19]. Furthermore,the American Diabetes Association (ADA) and European Association for the Study of Diabetes (EASD) recommends SGLT2 inhibitors early in the management of pacient with DM2 [20]. During the last few years, another promosing medication, named finerenone, has entered the medical market. It was found by at least two clinical trials, FIDELIO-DKD and FIGARO-DKD, that finerenone, a nonsteroidial selective mineralocorticoid receptor antagonist, reduced cardiorenal risk [21,22]. Finerenone was authorized by the FDA in 2021 and European Society of Cardiology guidelines recommend considering using the pharmaceutical agent in the treatment of heart failure with mildly reduced ejection fraction [23].”
Comment 2: The importance of therapy in anemia is missing!
Response:
We explain the effects of SGLT2 inhibitors introducing a paragraph and a figure: Figure 6.
„Some studies have shown that SGLT2 inhibitors are able to reduce ferritin, transferrin desaturation and hepcidin activity by improving iron absorption from the digestive tract and its mobilization from tissue stores and macrophages [219,220]. The advantage would be that SGLT2 inhibitors allow an increase in hemoglobin levels without administering iron supplements [221], an advantage also observed in patients with cardiorenal syndrome [223]. Docherrty K.F et al [223] studied this effect in the DAPA-HF trial, where, the administration of Dapagliflozin for 12 months in patients with heart failure NYHA classes II-IV and iron deficiency induced changes in the mean values ​​of hepcidin from 24.3 ng/mL to 17.2 ng/mL and ferritin from 158.2 ng/mL to 130 ng/mL. Similar results were observed with Canagliflozin in the CREDENCE trial in diabetic patients with CKD [224] and in the post-hoc analysis of the DELIGHT trial including similar cohorts of patients where Dapagliflozin, in addition to the previous findings, also produced an increase in erythropoietin to a mean of 82.9 pg/mL, compared to placebo (79.8 pg/mL) [225]. Murashima M et al [226] demonstrate that the administration of SGLT2 inhibitors improved anemia in diabetic patients with eGFR 15-30 mL/min/1.73 m2. Due to the low production of EPO, the increase was smaller in patients with eGFR ≤ 15 mL/min/1.73 m2 than in those with eGFR above 60 mL/min/1.73 m2. A study of the IRONMAN trial showed a comparatively greater increase in hemoglobin if an SGLT2 inhibitor was combined with intravenous administration of ferric derisomaltose [227].”
Comment 3: Supplement with effects on lipid profile and diastolic blood pressure!
Response: We supplemented with informations about diastolic BP and lipid profile influences produced by SGLT2 inhibitors, introducing the following paragraphs:
„Some studies have shown that SGLT2 inhibitors are able to reduce diastolic blood pressure (BP) but without affecting heart rate [126-130], with the most intense effect belonging to Canagliflozin [126]. Empagliflozin significantly reduced both systolic and diastolic BP only in patients with DM2 [131]. In a study that included 75 patients who were administered Canagliflozin 10 mg in a single dose per day, for 24 months, a reduction in systolic and diastolic BP was observed in diabetic patients but not for non-diabetic ones [132]. There is a hypothesis according to which the decrease in systolic and diastolic BP after the administration of SGLT2 inhibitors is due to the sympathetic modulation that they may exert on specific brain areas subject to autonomic control [133,134].”
„In a study [132], decreasing in total cholesterol concentration were observed after Dapagliflozin administration in patients with CKD, regardless of DM stage, but without significant changes in TGL, LDL, and HDL. There were only significant decline in LDL for patients with DM and high BP, but not to those without DM and high BP [132].”
Comment 4: Enough redundant text. Some parts should be converted into tables and visual representations. It has potential to be improved.
Response: We eliminated repetitions, removed the reduntant text and introduced 2 additional schemes having the new numbering as Figure 4 and Figure 6.
Reviewer 3 Report
Comments and Suggestions for Authors
This manuscript provides an overview of SGLT2 inhibitors, including structure, mechanism of action and non-diabetes related potential therapeutic applications. The topic is pertinent, and the scope of the paper is suitable for publication in the International Journal of Molecular Sciences, but some significant revision is necessary to bring the work up to standard and to update it to include the most recent advances in the field.
1- Revise the Introduction statistics on the prevalence of diabetes mellitus. The article quotes 537 million in 2021 and 828 million in 2022, and these numbers need to be verified using the most recent statistics published by the International Diabetes Federation. Also, provide more updated projections for 2024-2025 and check if the same method of reporting is used.
2- Please add to the discussion of SGLT2 inhibitors adverse events with recently published data. The following areas should be addressed:
- Expand on the euglycemic diabetic ketoacidosis section by including recent case reports and mechanistic data (2024).
- Include quantitative data on Fournier’s gangrene, amputations, and bone fracture risks as per recent meta-analyses.
- Better describe the role of dehydration and insulinopenia in the pathogenesis of euglycemic diabetic ketoacidosis.
3- Figures 4a and 4b should be revised to:
- Make them bigger and sharper for better legibility.
- Add more information about enzyme names and cofactors involved in the ketone body metabolism pathways.
- Include some quantitative flux information where recent metabolic studies exist.
- Use consistent chemical nomenclature in all of the figures.
4- Update cardiovascular protection mechanisms to include recent research:
- Discussion of differential effect of SGLT2 inhibitors vs. exogenous ketones on heart metabolism.
- Inclusion of mitochondrial biogenesis and autophagy's role in cardioprotection, including supporting evidence from recent (2023-2024) studies.
-Address the "controversy" of ketogenesis being the primary mechanism of cardiovascular benefit.
5- Please strengthen the literature review by:
- More references from 2024 for potential therapeutic uses
- Discuss clinical trials in heart failure and chronic kidney disease
- Add current structure-activity relationship studies from 2023-2024
- Ensure all cited sources are accessible and verifiable.
Comments on the Quality of English Language
Some Language Issues need to be revised:
- Fix the grammar in lines 63, 142, 276
- Improve sentence structure in the section for NAFLD (409-454)
- Correctly abbreviate all abbreviations the first time they are used and then standardize them.
Some Content Issues:
- Eliminate repetition in certain parts of the document, such as anti-inflammatory benefits
Author Response
Comment 1: Revise the Introduction statistics on the prevalence of diabetes mellitus. The article quotes 537 million in 2021 and 828 million in 2022, and these numbers need to be verified using the most recent statistics published by the International Diabetes Federation. Also, provide more updated projections for 2024-2025 and check if the same method of reporting is used.
Response:
We revised, corrected and completed statistics in way suggested by the referee:
„According to the International Diabetes Federation (IDF), there is a growing trend in the prevalence of DM, with 537 million cases estimated in 2021 and a forecast of increasing prevalence to 783 million by 2045 [7-9]. In particular, based on statistical data provided by IDF, the prevalence of diabetes is expected to affect 643 million people by 2030 [10]. Actually, it is estimated that of all patients with DM, 96% have DM2 [11].The related informations suggests that diabetes has reached pandemic proportions, impacting over half billion people worldwide [7].China remains the country with the highest diabetes rate, having approximately 140 million diabetic patients aged between 20-79 years old [12]. In a 2024 report, United States Centers for Disease Control showed significant statistics indicating that 14,7% of the adult population suffers from diabetes [13]. One of the biggest concerns in the medical field is undiagnosed diabetes. Several studies reveal that approximately 50% of all people with diabetes are undiagnosed [14]. Healthcare expenses associated with diabetes treatment are significantly higher than those for patients without diabetes [15]. The estimated diabetic-related costs for 2021 were 996 bilion USD and projected to reach 1,054 billion USD by 2045, with a major socioeconomic impact [7].”
Comment 2:
Please add to the discussion of SGLT2 inhibitors adverse events with recently published data. The following areas should be addressed:
- Expand on the euglycemic diabetic ketoacidosis section by including recent case reports and mechanistic data (2024).
- Include quantitative data on Fournier’s gangrene, amputations, and bone fracture risks as per recent meta-analyses.
- Better describe the role of dehydration and insulinopenia in the pathogenesis of euglycemic diabetic ketoacidosis.
Response:
- We expand the euglycemic diabetic ketoacidosis section by including recent case reports and mechanistic data (in addition to what exists or was introduced during the revision), published in 2024 as required by the referee:
„The factors that enhance EDKA appearance in the context of the use of SGLT2 inhibitors, are represented by the processes that enhance the ketone bodies synthesis (anorexia, ketogenic diet), dehydration (alcohol consumption or diuretic drugs), processes that increase oxidative stress and cause discharges of catecholamines and corticosteroids (infections, postoperative and intraoperative stress, stroke) [105-108]. Recent case reports showed EDKA in patients treated with gliflozins in different contexts associated with the factors described above. An 83-year-old patient with DM2 treated with empagliflozin was admitted to the emergency department with intraparenchymal cerebral hemorrhage in the left occipital lobe attributed to uncontrolled high blood pressure. The patient presented with metabolic acidosis but without dehydration. The authors believe that EDKA occurred in the context of oxidative stress and proinflammatory status generated by vascular and cerebral lesions that led to an imbalance in the glucagon/insulin ratio [109]. Another case of EDKA was in a 90-year-old patient with DM2 also treated with empagliflozin who presented with a urinary tract infection associated with anionic gap metabolic acidosis and with a history of ischemic cardiomyopathy and multiple myeloma in remission. Here, the anionic gap persisted 2 days after the start of insulin therapy [110]. Three other cases of EDKA in adult patients (68, 66 and 55 years old) with type 2 diabetes treated with SGLT2 inhibitors have been reported. Postoperative stress and lack of food intake have played an important role in the generation of ketoacidosis [111]. Moreover, other authors have described interesting cases in patients with DM2 who had an SGLT2 inhibitor in the therapeutic regimen, diagnosed with EDKA in the context of food deprivation [112], infections such as pulmonary [113], genital abscess [113], sepsis [114] or intraoperatively in the case of a craniotomy performed for tumor resection [116]. An atypical case was of an 83-year-old patient without DM where Dapagliflozin was administered for the treatment of heart failure. The precipitating factor that contributed to the development of EDKA was prolonged anorexia for 2 consecutive days of a vertebral and rib fracture. C-peptide values ​​were within normal limits, excluding damage to β pancreatic cells. Here, starvation generated an increased synthesis of ketone bodies in the context of glucose elimination, which had an important contribution even though insulin secretion was normal [117].”
- Recent meta-analyses including data on Fournier’s gangrene, amputations, and bone fracture risks were introduced.
„In a meta-analysis study, the results of 31 randomized controlled trials dedicated to peripheral artery disease and 15 for amputations were investigated. Here, slightly increased risks for amputations and peripheral artery disease were identified among subjects who used SGLT2 inhibitors compared with those who received antidiabetic drugs from other drug classes. An increased risk was observed only with the use of Canagliflozin [180]. A data analysis including some trials: EMPA-KIDNEY (containing patients with DM2+CDK), EMPA-REG OUTCOME (containing patients with DM2+/-CVD) and EMPEROR (containing patients with heart failure +/-DM2) showed that Empagliflozin does not present an increased risk of amputations, bone fractures and Fournier's gangrene (FD) [185]. On the other hand, the Canagliflozin Cardiovascular Assessment Study (CANVAS) reveals a twofold risk of lower limb amputations [186,187]. Nani A et al [211] in a systematic review performed a meta-analysis that included 42 randomized trials where amputations were described in 34 trials while lower limb fractures in 33 trials. The highest risk was observed with Canagliflozin, followed by Dapagliflozin and Ertugliflozin while Empagliflozin and Tofogliflozin were not accompanied by amputations. In contrast, all SGLT2 inhibitors presented a risk of bone fractures [186]. Other meta-analyses that evaluated the risk of lower limb amputation and bone fractures did not associate the incidence of these events with SGLT2 inhibitors, the values ​​being close to those of the control group [188] while others associate gliflozins with an increased frequency of both bone fractures and lower limb amputations [187,189].”
„Another concern is dedicated to FG, a infectious necrotizing fasciitis of the perineum and genitals in a fulminant form, found associated with SGLT2 inhibitors in context of glycosuria [233,234]. Although no exact explanation for the occurrence of FG could be found [234], there are studies where there is no correlation between FG and gliflozins [235]. According to FDA and Medicines and Healthcare Products Regulatory Agency, a special attention should be payed for FG for gliflozin users [235]. In a data analysis, involved 78 patients hospitalized for FG who required surgical reconstruction, of which 41% had DM, none of them were treated with SGLT2 inhibitors [235]. Despite these findings, cases of FG associated with SGLT2 inhibitors have been reported in the literature. A literature review that included data from 12 patients, highlighted 5 cases determined by Empagliflozin, 5 by Dapagliflozin and 2 by Canagliflozin [233]; Another review identified 13 case reports of FG, 8 of which were caused by Empagliflozin, the rest by Empagliflozin and Canagliflozin [236]. Additional new cases have been reported in the last year [237,238]. In the event of FG, treatment with SGLT2 inhibitor should be stopped and not resumed after healing [233]. Considering these aspects, even though FG is a rare event, should not be ignored in patients treated with SGLT2 inhibitors.”
- We describe better the role of dehydration and insulinopenia in the pathogenesis of euglycemic diabetic ketoacidosis as requested by the referee and indroduced a figure:
” The increase in the glucagon/insulin ratio in favor of glucagon enhances lipolysis with massive release of free fatty acids and synthesis of ketone bodies [91,92]. The lower the insulin production, the greater the risk of lipolysis and ketogenesis. Volume depletion combined with hypoinsulinemia are the triggering factors for ketoacidosis, activating the compensatory release of catecholamines and glucocorticoids with an even more pronounced increase in lipolysis in adipose tissue [93-95]. For this reason, in DM1, due to the absolute insulin deficiency with the compensation described above, the FDA does not recommend the use of SGLT2 inhibitors due to the major risk of producing EDKA, while the European Commission has allowed the use of Dapagliflozin and Sotagliflozin in patients with DM1 and body mass index over 27 kg/m2 since 2019 [96,97].”
„The administration of SGLT2 inhibitors will increase glycosuria with the reduction to normalization of blood glucose levels, making EDKA difficult to diagnose [92] (Figure 4). Also, at low blood glucose levels, the need for insulin cannot be accurately assessed [79]. In general, dehydration and hypoinsulinemia, each alone, are necessary but not sufficient to produce acidosis, but together they can generate EDKA. When insulinopenic rats are treated with Dapagliflozin, a 70% increase in adipose tissue lipolysis and ketogenesis results, producing EDKA[93]. Even though EDKA is a rare adverse effect, it has been described. In some studies, EDKA was an event observed only in diabetic patients treated with SGLT2 inhibitors [101]; in fact, in a group of patients receiving gliflozins, 56.3% suffered an episode of EDKA compared with those receiving other antidiabetic treatments (incidence of 2.6%) [102]. On the other hand, EDKA is 3.7 times more likely to occur during SGLT2 inhibitors treatment comparative to other drug classes [103].”
Comment 3:
- Figures 4a and 4b should be revised to:
- Make them bigger and sharper for better legibility.
- Add more information about enzyme names and cofactors involved in the ketone body metabolism pathways.
- Include some quantitative flux information where recent metabolic studies exist.
- Use consistent chemical nomenclature in all of the figures.
Response:
The format of the final form will provide a better quality and large figures, because we will upload separatelly the all figures.
Cofactors were introduced in figure 4a and 4b where they were missing; The names of the enzymes were modified on the schemes and/or on the figure captions and we introduced the improved chemical names modifying some abreviations and their explanation in figure captions. We include some citations [196,197,146] to explain NAD+/NADH+H+ ratio
Comment 4:
Update cardiovascular protection mechanisms to include recent research:
- Discussion of differential effect of SGLT2 inhibitors vs. exogenous ketones on heart metabolism.
- Inclusion of mitochondrial biogenesis and autophagy's role in cardioprotection, including supporting evidence from recent (2023-2024) studies.
-Address the "controversy" of ketogenesis being the primary mechanism of cardiovascular benefit.
Response:
Differential effects of SGLT2 inhibitors vs exogen ketones on heart metabolism were disscused:
„It is known that patients with heart failure and reduced ejection fraction have a high consumption of β-hydroxybutyrate [148]. Starting from the finding made by Nielsen et al [149] on the improvement of cardiovascular hemodynamics with an 8% increase in ejection fraction and 40% in cardiac output after exogenous administration of β-hydroxybutyrate [149], the benefit of exogenous administration of ketone bodies versus SGLT2 inhibitors was investigated [144,145]. Dapgliflozin was found to increase ketone body consumption, increasing myocardial ketolysis by 50%, reducing pyruvate oxidation, increasing plasma concentration of free fatty acids produced by lipolysis and directing them towards ketogenesis but without significantly influencing β-oxidation. A decrease in the NADH+H+/NAD+ ratio, reduction of oxidative stress, all materialized by the improvement of ejection fraction after 3 months, was also found. In comparison, exogenous β-hydroxybutyrate supplementation accelerates myocardial ketolysis with reduced β-oxidation of fatty acids but without altering glucose uptake and pyruvate oxidation. It also produces a decrease in free fatty acid uptake and an increase in the NADH+H+/NAD+ ratio [144,145].”
- The mitochondrial biogenesis and autophagy's role in cardioprotection, including supporting evidence from 2023 sau more recent studies was introduced.
„Mitophagy, an important process in maintaining myocardial cell function, can be activated in a ubiquitin-dependent or -independent manner. The most important pathway mediated by the activation of the kinase-ligase enzyme system is the PTEN-induced kinase 1 (PINK1)-Parkin pathway [165,166]. On the other hand, this process must be in balance with mitochondrial biogenesis to ensure efficient myocardial function. SGLT2 inhibitors are able to promote cardiomyocyte mitochondrial integrity by regulating both mitophagy and mitochondrial biogenesis [167]. Yang et al [166] demonstrate the involvement of Canagliflozin in both mitophagy and biogenesis. Mitophagy is promoted by the activating action of Canagliflozin on the PINK1-Parkin pathway. In intact mitochondria, PINK1 is transported to the inner mitochondrial membrane and protease-degraded [165,168]. A decrease in membrane potential will make it impossible for PINK to transport, which will dimerize and then autophosphorylate, becoming active [166,169]. In this form, PINK1 recruits Parkin and triggers the process of mitophagy, a mechanism activated by Canagliflozin in C57BL/6J mice with streptozocin-induced DM and fed with high fat diet (HFD) [166]. In the same mice, Canagliflozin improved mitochondrial biogenesis by upregulating the peroxisome proliferator-activated receptor G coactivator 1-α (PGC-1α)-mitochondrial transcription factor A (TFAM) pathway, a pathway with reduced expression under hyperglycemic conditions [166]. On the other hand, sirtuin-1 (SIRT-1) activated by SGLT2 inhibitors can directly induce autophagy by deacetylating autophagy-related genes [170]. Empagliflozin upregulates AMP-activating protein kinase (AMPK) by promoting its phosphorylation, the form in which it activates PGC-1A-TFAM [136,171-173]. Empagliflozin can reduce, on the one hand, the excessive upregulation of dynamin related protein 1 (Drp1) involved in mitochondrial fission and downregulated in diabetic Sprague-Dawley rats [172]; on the other hand, it attenuates the depletion of mitochondrial fusion proteins mitofusin 1 (Mfn1), Mfn2 and optic atrophy 1 (OPA1), proteins that are downregulated in mitochondrial alterations, and thus, lowering these alterations [172,173]. Empagliflozin can reverse myocardial microvascular damage resulting from DM by inhibiting mitochondrial fission mediated by AMPK [172]. Similar phenomena determined by Dapagliflozin have been observed in skeletal muscle tissue in Sprague-Dawley rats with induced DM2 [174].”
- We introduced a discussion involving "controversy" of ketogenesis being the primary mechanism of cardiovascular benefit:
„In general, cardiac muscle has an intense metabolism using fatty acids as a preferential energy substrate, which it β-oxidizes to produce ATP. Other alternative substrates are glucose and ketone bodies. In heart failure, the ability of the myocardium to β-oxidize fatty acids is reduced. Under conditions of satisfactory external nutritional intake, glucose becomes preferentially used under the action of insulin, while in the case of low-carbohydrate diets or starvation, fatty acids and then ketone bodies serve as the main energy source [140,141]. Ketogenic and carbohydrate-restricted diets as well as prolonged fasting have demonstrated their effectiveness in reducing body weight, blood pressure and blood glucose levels [142]. The other side of exogenous ketone bodies is the possibility of long-term alterations in cardiac function [143]. Ketogenic diet and administration of β-hydroxybutyrate reduce β-oxidation of fatty acids and their uptake by myocytes [144,145]. An increase in myocardial acetyl-CoA with the use of exogenous β-hydroxybutyrate could be responsible for the inhibitory effect on β-oxidation of fatty acids by modifying the NADH+H+/NAD+ or acetyl-CoA/CoA-SH ratio [144]. Treatment of C57BL/6J mice, which were induced with ischemic heart failure, with glucose, palmitate and β-hydroxybutyrate and fed a ketogenic diet showed a 56% decrease in ejection fraction [143]. High circulating acetoacetate concentrations have been associated with a high number of deaths in patients with heart failure, without these values ​​being significantly influenced by stress. In addition, they have a potential harmful effect on the vessels, with increased LDL levels, arterial stiffness and functional endothelial damage after prolonged ketogenic diets [146]. These two sides of ketone bodies, beneficial and harmful, may be time-dependent, where short-term exogenous administration or intermittent ketogenic diets may present an advantage while long-term exposure to elevated ketone body concentrations may have potentially detrimental effects [146,147]”
Comment 5:
Please strengthen the literature review by:
- More references from 2024 for potential therapeutic uses
- Discuss clinical trials in heart failure and chronic kidney disease
- Add current structure-activity relationship studies from 2023-2024
- Ensure all cited sources are accessible and verifiable.
Response:
We introduced more references from 2024 for potential therapeutic uses
- We engaged a discussion related to clinical trials in heart failure and chronic kidney disease:
„Evidence from recent clinical trials reinforces the established benefits of SGLT2 in terms of management of patients with heart failure and CKD. A multicenter, randomized trial shows that after 12 weeks of treatment with dapagliflozin, an SGLT2 inhibitor, symptoms and physical status of patients with heart failure with preserved ejection fraction improved considerably [203]. According to the data obtained from a study involving 507 critically ill patients, even tough organ dysfunction was not significantly improved by the use of dapagliflozin, it was still well tolerated in this high-risk category [204]. A reference clinical study for the therapeutic effects of SGLT2 remains EMPA-Kidney, which confirmed the cardiorenal protective action of SGLT2 inhibitors,supporting their use in both primary prevention and treatment of CKD [205]. Findings from the DECLARE-TIMI 58 trial showed that dapagliflozin descrease eGFR, albuminuria and has a role key in the prevention of diabetic kidney disease [206,207].”
- New references refer to selectivity and structure relationship were introduced : 72,73 and 74
All cited sources are accessible and verifiable
Comment 6:
Comments on the Quality of English Language
Some Language Issues need to be revised:
- Fix the grammar in lines 63, 142, 276
- Improve sentence structure in the section for NAFLD (409-454)
- Correctly abbreviate all abbreviations the first time they are used and then standardize them.
Some Content Issues:
- Eliminate repetition in certain parts of the document, such as anti-inflammatory benefits
Response: We fixed the grammar errors, improved the sentence structure in NAFLD section, corrected abreviations and eliminated repetition related to anti-inflammatory benefits.
Reviewer 4 Report
Comments and Suggestions for Authors
This manuscript provides a systematic review of SGLT. The article is easy to read, however I have some questions. Please make clear.
- This manuscript also touches on the use of SGLT2 inhibitors, however there is little clinical research data provided. Please include not only basic data but also clinical research.
- The final chapter of 4.2.3 states that the use of SGLT2 inhibitors increases lower limb amputations. However, recent clinical studies have not shown an increase in lower limb amputations. Is this stated from mechanisms? Please consider a literature review, etc.
- Please mention any adverse events caused by SGLT2 inhibitors (such as cases in which they should not be used).
- There are some errors, such as the abbreviation for renin-angiotensin-aldosterone system (RAAS) appearing twice. Please check.
Author Response
Comment 1: This manuscript also touches on the use of SGLT2 inhibitors, however there is little clinical research data provided. Please include not only basic data but also clinical research.
Response: We have included clinical research data throughout the manuscript as requested by the reviewer.
Comment 2: The final chapter of 4.2.3 states that the use of SGLT2 inhibitors increases lower limb amputations. However, recent clinical studies have not shown an increase in lower limb amputations. Is this stated from mechanisms? Please consider a literature review, etc.
Response:
We introduced a literature review with controversial results regarding risk of lower limb amputations, some results being drug-related:
„In a meta-analysis study, the results of 31 randomized controlled trials dedicated to peripheral artery disease and 15 for amputations were investigated. Here, slightly increased risks for amputations and peripheral artery disease were identified among subjects who used SGLT2 inhibitors compared with those who received antidiabetic drugs from other drug classes. An increased risk was observed only with the use of Canagliflozin [180]. A data analysis including some trials: EMPA-KIDNEY (containing patients with DM2+CDK), EMPA-REG OUTCOME (containing patients with DM2+/-CVD) and EMPEROR (containing patients with heart failure +/-DM2) showed that Empagliflozin does not present an increased risk of amputations, bone fractures and Fournier's gangrene (FD) [185]. On the other hand, the Canagliflozin Cardiovascular Assessment Study (CANVAS) reveals a twofold risk of lower limb amputations [186,187]. Nani A et al [211] in a systematic review performed a meta-analysis that included 42 randomized trials where amputations were described in 34 trials while lower limb fractures in 33 trials. The highest risk was observed with Canagliflozin, followed by Dapagliflozin and Ertugliflozin while Empagliflozin and Tofogliflozin were not accompanied by amputations. In contrast, all SGLT2 inhibitors presented a risk of bone fractures [186]. Other meta-analyses that evaluated the risk of lower limb amputation and bone fractures did not associate the incidence of these events with SGLT2 inhibitors, the values ​​being close to those of the control group [188] while others associate gliflozins with an increased frequency of both bone fractures and lower limb amputations [187,189].”
Comment 3: Please mention any adverse events caused by SGLT2 inhibitors (such as cases in which they should not be used).
Response:
We introduced a comment related to the adverse events caused by SGLT2 inhibitors in which they should not be used as required by the referee
” On the other hand, EDKA is 3.7 times more likely to occur during SGLT2 inhibitors treatment comparative to other drug classes [103]. Because EDKA is a life-threatening condition, SGLT2 inhibitor therapy should be discontinued once it occurs [104]. In the event of an episode of EDKA, it is not recommended to restart the treatment with SGLT2 inhibitors after the patient has been stabilized because there is an increased risk of recurrence [105].”
„In the event of FG, treatment with SGLT2 inhibitor should be stopped and not resumed after healing [233].”
Comment 4: There are some errors, such as the abbreviation for renin-angiotensin-aldosterone system (RAAS) appearing twice. Please check.
Response: The abbreviation for renin-angiotensin-aldosterone system (RAAS) which appearing twice was corrected.
Round 2
Reviewer 2 Report
Comments and Suggestions for Authors
No
Reviewer 3 Report
Comments and Suggestions for Authors
No comments
Thank you for addressing all previous points.